# Structural basis for phage-mediated activation and repression of bacterial DSR2 anti-phage defense system

Jun-Tao Zhang[1,6], Xiao-Yu Liu[1,6], Zhuolin Li[1,6], Xin-Yang Wei[1], Xin-Yi Song[1], Ning Cui[1,2,3], Jirui Zhong[4,5], Hongchun Li [4,5] & Ning Jia [1,2,3] ✉

Silent information regulator 2 (Sir2) proteins typically catalyze NAD+-dependent protein deacetylation. The recently identified bacterial Sir2 domain-containing protein, defense-associated sirtuin 2 (DSR2), recognizes the phage tail tube and depletes NAD+ to abort phage propagation, which is counteracted by the phage-encoded DSR anti-defense 1 (DSAD1), but their molecular mechanisms remain unclear. Here, we determine cryo-EM structures of inactive DSR2 in its apo form, DSR2–DSAD1 and DSR2–DSAD1–NAD+, as well as active DSR2–tube and DSR2–tube–NAD+ complexes. DSR2 forms a tetramer with its C-terminal sensor domains (CTDs) in two distinct conformations: CTD$^{closed}$ or CTD$^{open}$. Monomeric, rather than oligomeric, tail tube proteins preferentially bind to CTD$^{closed}$ and activate Sir2 for NAD+ hydrolysis. DSAD1 binding to CTD$^{open}$ allosterically inhibits tube binding and tube-mediated DSR2 activation. Our findings provide mechanistic insight into DSR2 assembly, tube-mediated DSR2 activation, and DSAD1-mediated inhibition and NAD+ substrate catalysis in bacterial DSR2 anti-phage defense systems.

Sir2 (silent information regulator 2) proteins are an ancient family of NAD+-dependent enzymes that are widely distributed across all three domains of life and typically catalyze protein deacetylation or ADP-ribosylation using NAD+ as a cofactor[1–3]. They typically require peptide substrates containing acetyl-lysine and the cofactor NAD+ for catalysis, leading to the deacetylation of acetyl-lysine and the release of nicotinamide (NAM) and 2′-O-acetyl-ADP-ribose (ADPR) as reaction products. In eukaryotes, the Sir2 proteins are involved in various biological processes, including transcriptional repression, recombination, the cell-division cycle, microtubule organization, cellular responses to DNA-damaging agents, and even life span extension[2]. Eukaryotic Sir2 proteins have emerged as promising pharmacological targets due to their associations with age-related human diseases,

including cancer, neurodegeneration, cardiovascular diseases, dyslipidemia, obesity, and diabetes, as well as pathological conditions such as bacterial infections, human immunodeficiency virus type-1 (HIV-1) transcription, and inflammatory responses[4]. In prokaryotes, bacterial Sir2 proteins participate in transcription, protein translation, carbon and nitrogen metabolism, and virulence[3,5,6].

In contrast to the reported Sir2 proteins that typically function individually, recent bioinformatic analysis has uncovered a variety of Sir2 domain-containing proteins involved in bacterial anti-phage defense systems[7,8], including antiviral ATPase/NTPase of the STAND superfamily (AVAST)[9], Thoeris[10–13], prokaryotic argonautes (pAgo)[14], and defense-associated sirtuin 1 (DSR1) and DSR2[8]. These Sir2 domain-containing proteins typically consists of a Sir2 domain serving as an

[1]Department of Biochemistry, School of Medicine, Southern University of Science and Technology, 518055 Shenzhen, China. [2]Shenzhen Key Laboratory of Cell Microenvironment, Guangdong Provincial Key Laboratory of Cell Microenvironment and Disease Research, Southern University of Science and Technology, 518055 Shenzhen, China. [3]Key University Laboratory of Metabolism and Health of Guangdong, Institute for Biological Electron Microscopy, Southern University of Science and Technology, 518055 Shenzhen, China. [4]Research Center for Computer-Aided Drug Discovery, Shenzhen Institute of Advanced Technology, Chinese Academy of Sciences, 518055 Shenzhen, China. [5]Biomedicial Department, University of Chinese Academy of Sciences, 100049 Beijing, China. [6]These authors contributed equally: Jun-Tao Zhang, Xiao-Yu Liu, Zhuolin Li. ✉e-mail: jian@sustech.edu.cn

NADase effector, associated with a sensor domain that can detect the invading phages. Approximately half of the identified short pAgos are associated with Sir2 domains[15]. Upon RNA-guided detection of invading phage or plasmid DNA by a short pAgo, the Sir2 domain of *Geobacter sulfurreducens* short pAgo and the associated Sir2-APAZ (SPARSA) depletes essential NAD⁺, resulting in cell death[14]. In the Thoeris defense systems, the C-terminal Smf/DprA-LOG (SLOG)-like sensor domain of ThsA detects isomers of the signaling cyclic adenosine diphosphate–ribose (cADPR), which is produced upon phage infection, after which its N-terminal Sir2 domain degrades NAD⁺ [10–13]. In addition to recognizing invading phage nucleic acids or secondary signaling molecules produced upon phage invasion, the Sir2 domain-containing protein DSR2 from *Bacillus subtilis* defends against the phage SPR by directly recognizing the phage tail protein, thereby triggering the depletion of essential cellular NAD⁺ via its Sir2 domain[8]. To counteract this bacterial DSR2-mediated defense system, some phages carry genes encoding anti-DSR2 proteins known as DSR anti-defense 1 (DSAD1), which bind directly to and repress the NADase activity of the Sir2 domain[8].

Despite extensive identification of Sir2 domain-containing proteins, the molecular mechanisms underlying their function in anti-phage defense systems remain poorly understood. Here, using cryogenic electron microscopy (cryo-EM) combined with in vitro biochemical and in vivo analysis, we elucidated the molecular mechanisms underlying the phage tail tube-mediated activation and phage-encoded anti-DSR2 protein DSAD1-mediated inhibition of DSR2. These findings deepen our understanding of these widely distributed bacterial Sir2-domain–containing anti-phage defense systems and provide insights into the ongoing evolutionary interactions between bacteria and phages.

## Results

### DSR2 assembles into a head-to-head tetramer

To investigate the molecular mechanism by which the DSR2 anti-phage defense system functions, we co-produced *B. subtilis* DSR2 with the tail tube protein of phage SPR in *Escherichia coli* cells. However, these proteins appeared to be toxic to the bacterial cells and resulted in cell death, supporting previous observations that the phage tail tube protein activates DSR2 in vivo[8] (Fig. 1a, b). To investigate how DSR2 proteins assemble, we produced *B. subtilis* DSR2 in *E. coli* cells and then purified it from the cell lysate. Size exclusion chromatography analysis revealed that DSR2 existed as an oligomer (Supplementary Fig. 1a).

We initially determined a 2.6-Å cryo-EM structure of DSR2 in its apo form with approximately 60% density observed, which facilitated model building. Subsequent 3D classification allowed us to obtain a 3.1-Å cryo-EM structure of the apo form of DSR2 containing all parts of DSR2 density (Fig. 1c–f and Supplementary Fig. 1b–i). The overall architecture of the DSR2 complex assembled in a tetrameric state, forming a 'dimer of dimers'. DSR2 contains an N-terminal Sir2 domain that serves as an NADase effector responsible for NAD⁺ hydrolysis, a C-terminal domain (CTD) that functions as a sensor for detecting invading phages (as discussed later), and a middle domain (MID) that connects the Sir2 effector and the CTD sensor (Fig. 1c). Four Sir2 domains are arranged in a central core, adopting a head-to-head arrangement, while the CTDs are positioned at the periphery of the complex. Protomers 1 and 2, as well as protomers 3 and 4, form a basic X-shaped dimer through interactions mediated by MID-to-MID swapping and Sir2-to-Sir2 interactions, resulting in a total buried interface of ~4200 Å² (Fig. 1f). The two basic X-shaped dimers are further organized in a head-to-head arrangement via Sir2-mediated interactions, with a buried interface area of ~2000 Å².

A structural comparison among the four protomers revealed that protomers 1 and 3, as well as protomers 2 and 4, share similar conformations, while protomers 1 and 2, as well as protomers 3 and 4, exhibit distinct conformations (Fig. 1g and Supplementary Fig. 2a).

When superimposing protomer 1 onto protomer 2, we observed similar conformations for Sir2 and MID with a respective root mean square deviations (RMSD) of 0.205 Å and 0.373 Å, but substantially conformational differences for the CTD with an RMSD of 3.763 Å (Supplementary Fig. 2b). Specifically, the CTD in protomers 1 and 3 adopts a closed conformation (CTD^closed), while that of protomers 2 and 4 takes on an open conformation (CTD^open) (Fig. 1d, h). These conformational differences contribute to the recognition of invading phages (see below).

The Sir2 domain of DSR2 adopts a well-conserved large Rossman fold for NAD⁺ binding, and a small helical module consisting of four α helices (α3–α6) (Supplementary Fig. 2c). Structural comparisons revealed that the Sir2 domain of DSR2 more resembles the Sir2 domain of ThsA (PDB 6LHX, Z score 21.4, RMSD of 4.3 Å over 224 atoms) (Supplementary Fig. 2d) responsible for NAD⁺ hydrolysis[11,12], than that of eukaryotic Sir2 proteins, such as the Sir2 protein Sir2Af1 from *Archaeoglobus fulgidus* (PDB 4TWI, Z score 15.0, RMSD of 2.8 Å over 190 atoms) (Supplementary Fig. 2e), which requires both an acetyl-lysine–containing peptide substrate and an NAD⁺ cofactor for catalysis[16]. We noticed that the Sir2 domain of DSR2 lacks a zinc-binding insertion motif typically present in Sir2 proteins, which is critical for maintaining the integrity of the catalytic site[1] (Supplementary Fig. 2e). The channel responsible for the binding of acetyl-lysine containing peptide substrates is blocked by the presence of α11 (Supplementary Fig. 2e), suggesting that the Sir2 domain within DSR2 cannot function as an NAD⁺-dependent deacetylase. Thus, our findings suggest that the tetrameric DSR2 should function as a tube-activated NAD⁺ hydrolase rather than an NAD⁺-dependent deacetylase.

### The monomeric form of phage SPR tail tube, but not its oligomeric form, activates the DSR2 NADase activity

Since co-expression of wild-type DSR2 (WT) with the tube protein is toxic to bacterial cells (Fig. 1a), we separately expressed and purified the wild-type DSR2 and the tail tube protein from phage SPR from the cell lysates for in vitro biochemical studies on the tube protein-mediated DSR2 activation. Size exclusion chromatography analysis indicated the presence of two distinct states for the tail tube protein: monomeric and oligomeric states (Fig. 2a). Only the monomeric form of the tail tube protein activated the NADase activity of DSR2 (Fig. 2b). To gain structural insights into how the monomeric phage tail tube protein activates DSR2, we purified the DSR2–tube complex by separately expressing the wild-type DSR2 and the tube protein in *E. coli* cells, and mixed the cell pellets before ultrasonication and purification. Affinity and size exclusion chromatography analysis revealed that the phage tail tube protein formed a stable complex with DSR2 (Fig. 2c). The purified DSR2–tube complex exhibited pronounced NADase activity (Fig. 2d), further demonstrating that binding of the tail tube protein to DSR2 activates its NAD⁺ hydrolysis activity in vitro (Fig. 2d). Mutation of the conserved residues N133 or H171 within the Sir2 domain to alanine decreased the NADase activity of DSR2 proteins (Fig. 2d and Supplementary Fig. 2f, g), a result that aligns well with previous in vivo studies[8], demonstrating that the Sir2 domain of DSR2 is responsible for NAD⁺ hydrolysis. Collectively, our findings demonstrate that the monomeric form of phage tail tube proteins, rather than their oligomeric form, activates the NADase activity of DSR2.

### The tail tube protein of phage SPR preferentially binds to the CTD^closed within DSR2 in protomers 1 and 3

To gain structural insights into how the monomeric rather than the oligomeric phage tail tube proteins activate the NADase activity of DSR2, we first purified the DSR2-tube complex by separate expression of DSR2 and tube proteins, followed by mixing their cell pellets for the subsequent ultrasonication, as coexpression of DSR2 and tail tube proteins are toxic to the bacterial cells. We then determined the cryo-EM structure of the DSR2–tube complex at 3.6-Å resolution (Fig. 2e–g,

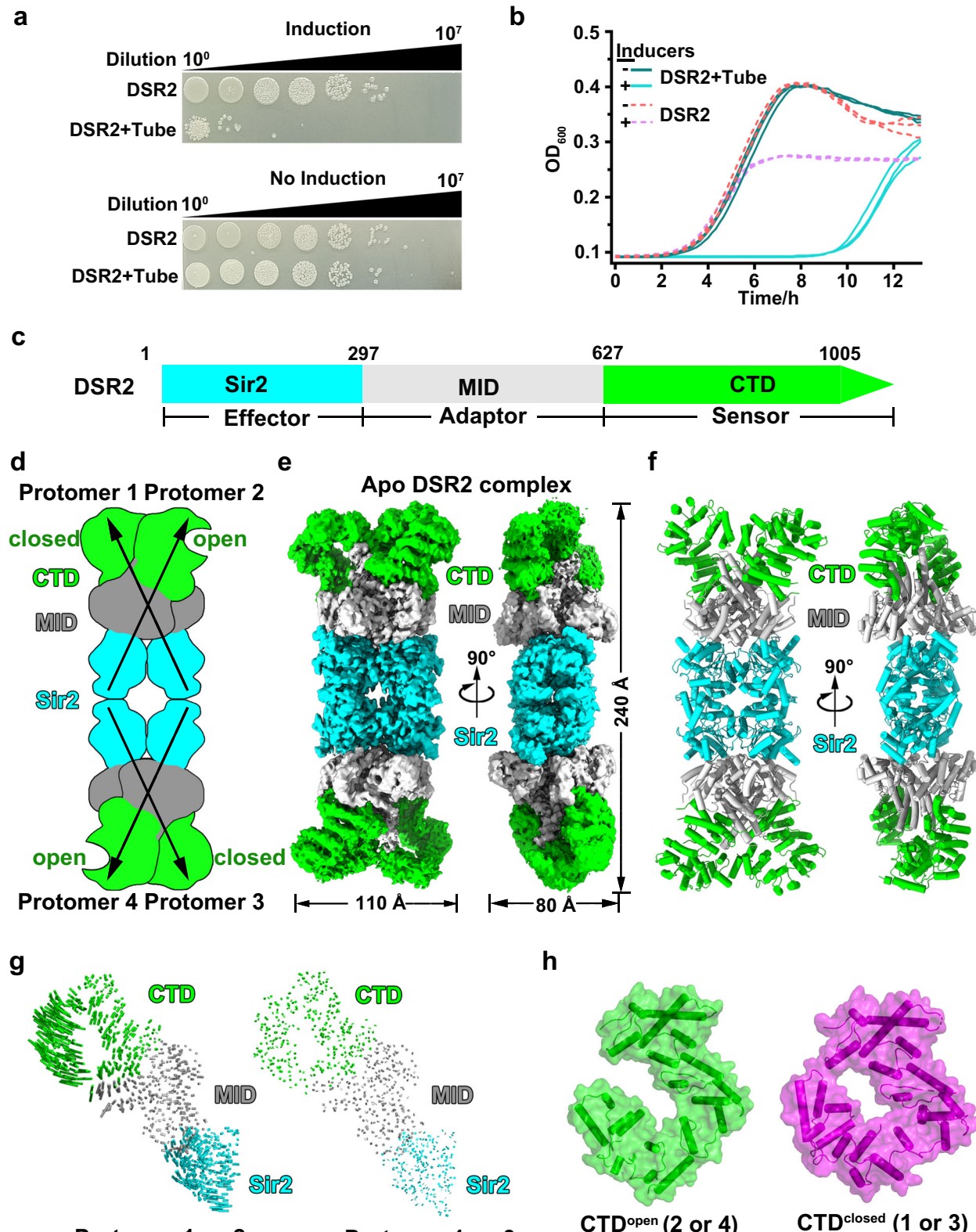

**Fig. 1 | Overall architecture of the apo DSR2 tetramer. a** Survival status of *E. coli* cells producing DSR2 alone or together with the phage tail tube proteins. **b** Growth curves of *E. coli* cells producing DSR2 alone or together with the phage tail tube protein. Protein production was induced by the addition of 0.5 mM IPTG and 1% L-Ara. The *E. coli* cells without induction were used as control. Curves represent three independent experiments. **c** Diagram of DSR2 showing all functional domains (NCBI protein accession: WP_029317421). **d-f** Schematic (**d**), surface (**e**) and ribbon (**f**) representations of the 3.1-Å cryo-EM structure of the tetrameric apo form of the DSR2 complex. **g** Structural comparison of Protomer 1 relative to Protomer 2 and Protomer 1 compared to Protomer 3. Vector length correlates with the domain movement scale. **h** Two different conformations for CTD in the protomers of tetrameric DSR2. Source data are provided as a Source Data file.

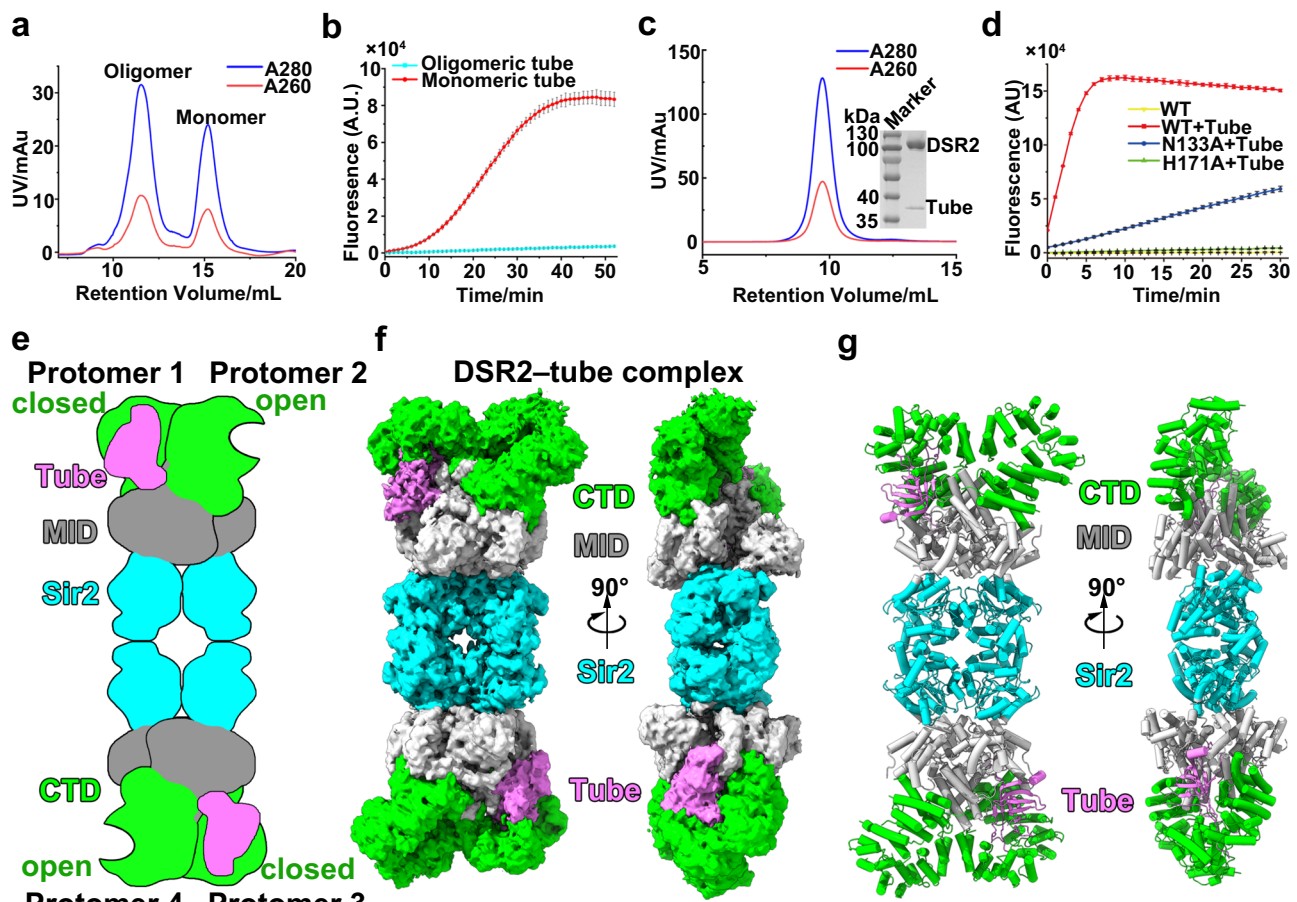

**Fig. 2 | Monomeric rather than oligomeric phage SPR tube proteins directly bind and activate the NADase activity of DSR2. a** Size exclusion chromatography profile of tail tube proteins from phage SPR. **b** NADase activity detected for 400 nM DSR2 in the presence of 2 μM monomeric or 2 μM oligomeric phage tubes proteins ($n = 3$ independent experiments; mean ± SD). **c** Size exclusion chromatography profile of the DSR2–tube protein complex. **d** NADase activity of WT DSR2, DSR2–tube, DSR2$^{N133A}$–tube and DSR2$^{H171A}$–tube complexes ($n = 3$ independent experiments; mean ± SD). **e–g** Schematic (**e**), surface (**f**) and ribbon (**g**) representations of the 3.6-Å cryo-EM structure of the DSR2–tube binary complex. Source data are provided as a Source Data file.

Supplementary Fig. 3a–c), with a DSR2: tail tube ratio of 4:2. Each monomeric tail tube binds to the CTD$^{closed}$ in protomers 1 and 3, with a buried interface of ~3000 Å². Given that protomers 3 and 4, respectively, exhibit structural similarity to protomers 1 and 2 (Fig. 1g, Supplementary Fig. 2a), we mainly focused on protomers 1 and 2 for the subsequent analyses. To elucidate the preference of phage tail tube proteins for binding to the CTD$^{closed}$ in protomers 1 and 3, we superimposed the tail tube–bound protomer of the DSR2·tube complex with either protomer 1 containing CTD$^{closed}$, or protomer 2 containing CTD$^{open}$ from the DSR2 apo form. We observed more pronounced conformational changes for CTD$^{open}$ of protomer 2 that are needed for binding to the tail tube, in contrast to the CTD$^{closed}$ in protomer 1 (Fig. 3a), suggesting that the pocket in CTD$^{closed}$ in protomers 1 and 3 is better configured for tail tube binding than in CTD$^{open}$ in protomers 2 and 4.

The tail tube protein of phage SPR consists of two separate domains, Domain1 (D1) and Domain2 (D2); we only observed densities corresponding to the D1 but not the D2, as modeled by Alphafold 2[17], suggesting the flexibility of D2 (Fig. 3b, Supplementary Fig. 3d). Structural analysis using Dali search revealed that the D1 in the phage SPR tail tube closely resembles D1 of the tail tube protein from phage YSD1 (Fig. 3b, c)[18]. D1 from SPR tail tube protein contains a conserved β-sandwich structure present in the tail tube protein of various long-tailed phages (Supplementary Fig. 3e), while the peripheral D2 exhibits diversity and is believed to play a role in phage tail stabilization or host surface attachment[18–20]. In phage YSD1, six tail tube proteins interacted

with each other to form the tail tube through interconnections of their conserved D1s, ultimately assembling into an antiparallel β-barrel structure (Fig. 3c). This structural analysis suggests that D1 in the tail tube protein of phage SPR contributes to the formation of the central hexametric ring within the tail tube structure.

D1 in the tail tube protein of phage SPR consists of a β-sandwich structure with two long loops extending from β2, β3 (loop1) and β6, β7 (loop 2), respectively (Fig. 3b). Loop 1 and loop 2, respectively, inserts into the CTD$^{closed}$ and MID of protomer 1 (Fig. 3d, Supplementary Fig. 3f). The β8 and β2 strands of the tail tube proteins, respectively, forms anti-parallel β-strand interactions with the β strands from the CTD$^{closed}$ of protomer 1 and MID of protomer 2 (Fig. 3d). To further highlight the importance of these two loops, we replaced residues in loop1 (residues Q34–K57, tube$^{ΔLoop1}$) or residues in loop 2 (residues F204–P216, tube$^{ΔLoop2}$) with a GS linker. Co-expression of DSR2 with the tube$^{ΔLoop1}$ did not exhibit notable toxicity to bacterial cells, whereas co-expression of DSR2 with the tube$^{ΔLoop2}$ remained cytotoxic (Fig. 3e). To verify the loss of the toxicity was not due to poor expression of tube$^{ΔLoop1}$ mutant, we purified the DSR2–tube$^{ΔLoop1}$ complex. Our results revealed that tube$^{ΔLoop1}$ mutant forms a stable complex with DSR2 in a molecular ratio similar to the wild type (Supplementary Fig. 3g, h). In agreement with our in vivo observations, the DSR2–tube$^{ΔLoop1}$ complex showed a significant drop in its NADase activity relative to the DSR2–tube complex (Fig. 3f), demonstrating the essential role of loop1 for DSR2 activation. Taken together, the above results showed that DSR2 recognizes the conserved D1 domain of the tail tube protein of

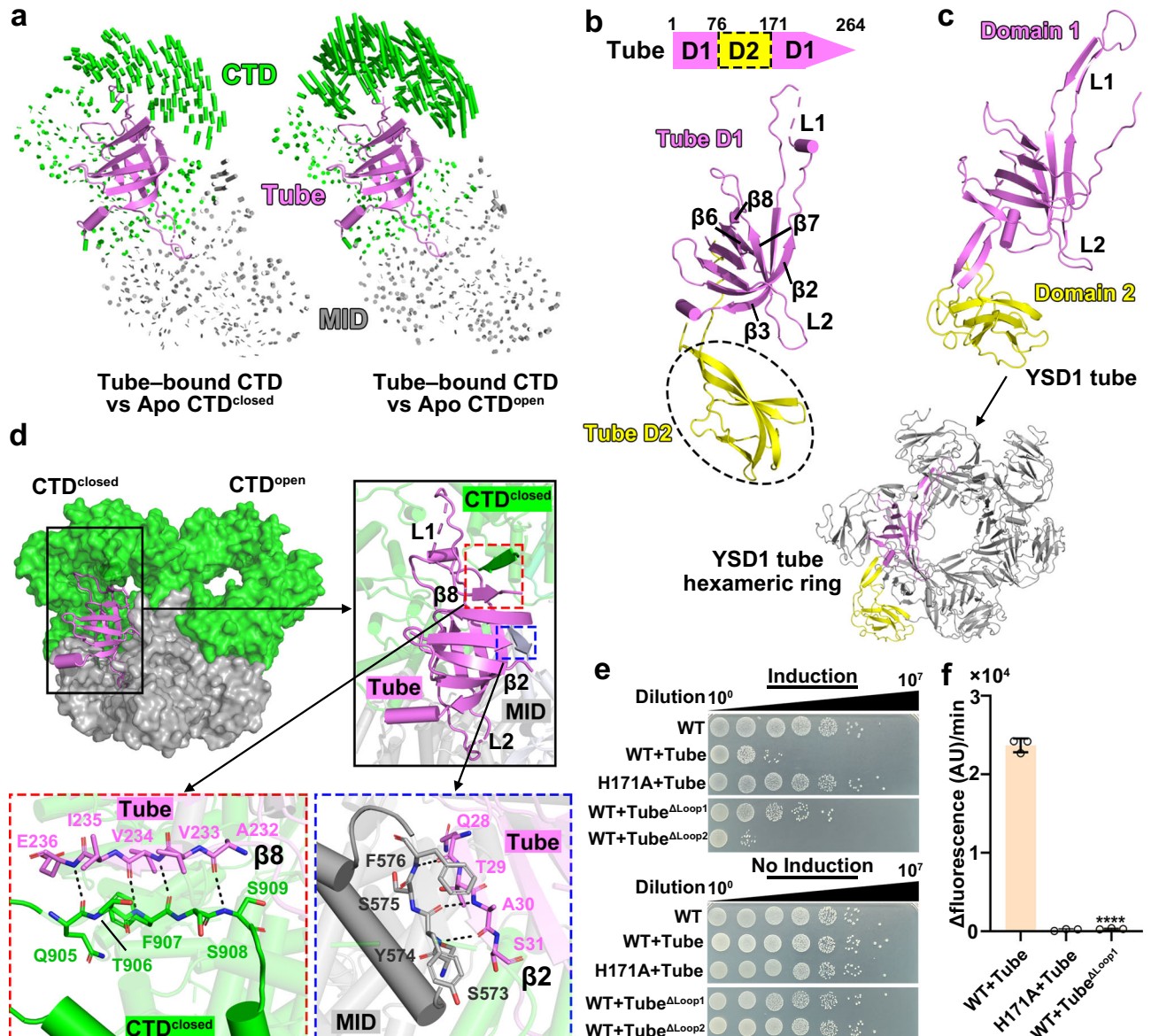

**Fig. 3 | Recognition of the tail tube protein from phage SPR tail tube by CTD^closed in protomers 1 and 3. a** Structural comparison of tube-bound CTD with CTD^closed (close conformation, left) and CTD^open (open conformation, right) of the apo form of DSR2. Vector length correlates with the scale of domain movement. **b** Diagram of domain organization (top) and ribbon representations (bottom) of monomeric tail tube protein from phage SPR. **c** Ribbon representations of the monomeric and hexameric form of the YSD1 tube protein (PDB 6XGR). **d** The β8 and β2 strands of

the tail tube protein of phage SPR respectively interact with CTD^closed domain of protomer 1 and the MID of protomer 2. **e** Survival status of *E. coli* cells co-producing DSR2 (WT or variants) and tube (WT or variants). **f** NADase activity of WT DSR2, DSR2^H171A–tube and DSR2–tube^ΔLoop1 complexes (n = 3 independent experiments; mean ± SD). The p-values were calculated using a one-sided Student's t test [****: p < 0.0001; p = 0.000000657]. Source data are provided as a Source Data file.

phage SPR by its CTD^closed in protomers 1 and 3. This recognition explains why only the monomeric tail tube proteins, but not their oligomeric form, activate the NADase activity of DSR2, as D1 becomes buried and inaccessible for DSR2 interaction once the tail tube ring structure has formed.

## Tetramerization is crucial for the activation of DSR2 NADase activity

DSR2 exists as a tetramer in solution regardless of the presence of the phage tail tube protein (Supplementary Fig. 1a). Within tetrameric DSR2, two DSR2 protomers (protomers 1 and 2; protomers 3 and 4) form a basic X-shaped dimer through interactions mediated by the MID-to-MID and Sir2^1-to-Sir2^2 or Sir2^3-to-Sir2^4 interface (Fig. 4a, black and yellow insets). These interactions involve four pairs of hydrogen-bonding interactions, between residues H606, Q610, and R613 in

protomer 1, and E560, T562, and N563 in protomer 2, for the MID-to-MID interaction (Fig. 4a, black inset), and two pairs of hydrogen-bonding interactions, between residues N202 and T206, for the Sir2^1-to-Sir2^2 or Sir2^3-to-Sir2^4 interfaces (Fig. 4a, yellow inset). These two basic dimers further combine in a head-to-head pattern via Sir2^1-to-Sir2^4 and Sir2^2-to-Sir2^3 interfaces (Fig. 4a, red inset), involving in four pairs of hydrogen bonds. Residues Y71 and D188 from the Sir2 domain in one protomer form direct hydrogen bonds with residues T257 and R233 from the Sir2 domain in the opposite protomer (Fig. 4a, red inset). Co-expression of either DSR2 double mutants DSR2^Q610A, R613A or DSR2^N202A, T206A with the tube proteins had a negligible effect on the DSR2 activation by the tail tube protein (Fig. 4b). However, the DSR2^Y71A, D188A double mutant but not DSR2^Y71A, DSR2^D188A single mutants significantly impaired DSR2 activation, resulting in no toxicity to bacterial cells when co-expressed with the tail tube protein (Fig. 4b,

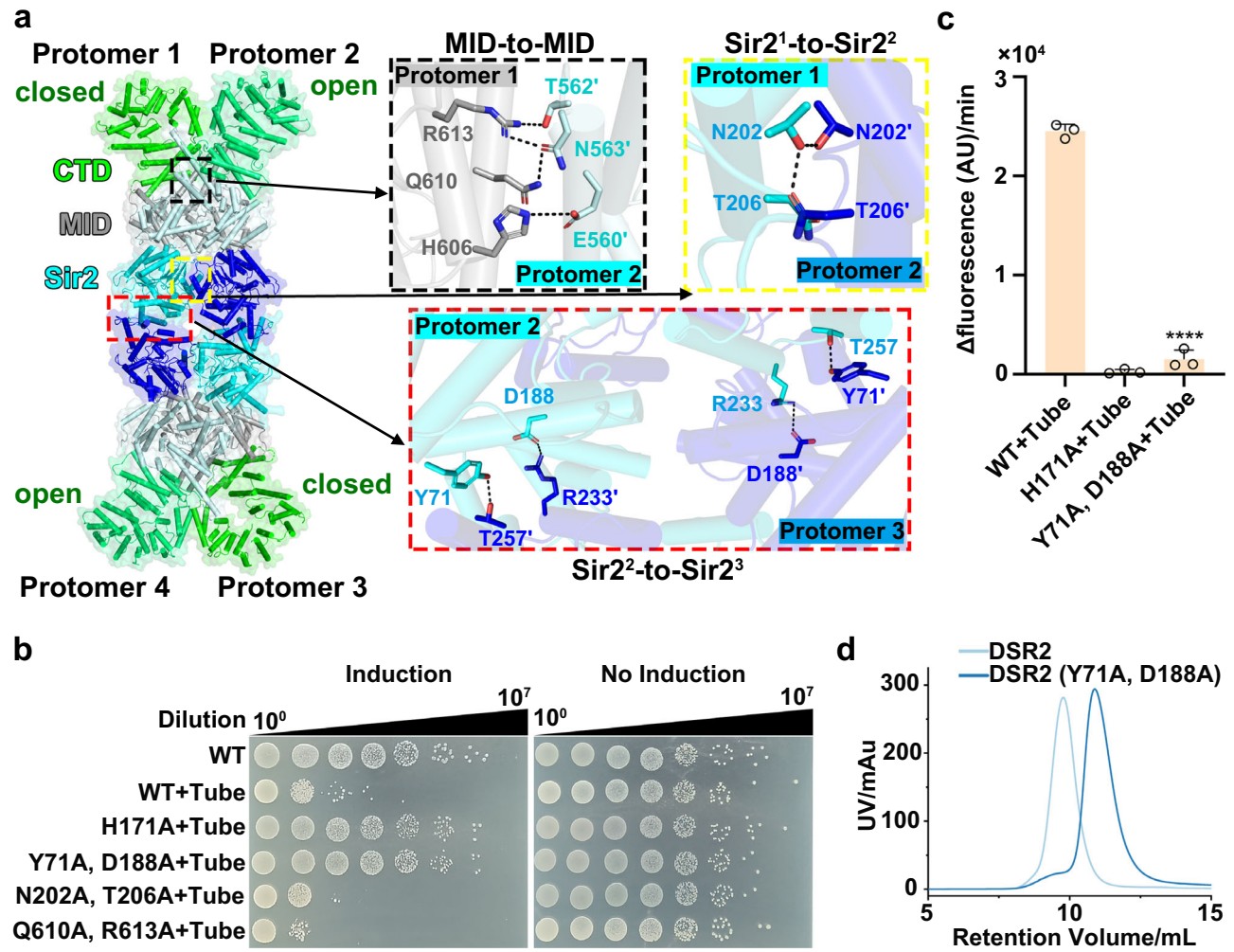

**Fig. 4 | Tetramerization of DSR2 is indispensable for its NADase activity.**
**a** Different interfaces contribute to the formation of the DSR2 tetramer. **b** Survival status of *E. coli* cells co-producing DSR2 (WT or variants) and tube proteins. **c** NADase activity of WT DSR2, DSR2^H171A–tube and DSR2^Y71A, D188A–tube complexes

($n = 3$ independent experiments; mean ± SD). The *p*-values were calculated using a one-sided Student's *t* test [****: $p < 0.0001$; $p = 0.00000221$]. **d** Size exclusion chromatography profiles of WT DSR2 and DSR2^Y71A, D188A. Source data are provided as a Source Data file.

Supplementary Fig. 3i), highlighting that the loss of activity in DSR2 activation requires the presence of double mutants. To verify our in vivo observations, we purified the DSR2^Y71A, D188A mutant in complex with the phage tail tube protein: this double mutant had much lower NADase activity than its WT counterpart (Fig. 4c). Size exclusion chromatography and mass photometry analysis revealed that the DSR2^Y71A, D188A mutant was smaller than wild-type DSR2, indicating that these mutations disrupted the tetrameric assembly of DSR2 (Fig. 4d, Supplementary Fig. 3j, k). These results highlighted the crucial role of residues Y71 and D188 within the Sir2 domain for DSR2 tetramerization and demonstrate the indispensable role of DSR2 tetramerization in the activation of its NADase activity.

### The NAD+ substrate binds to each Sir2 domain within the tetrameric DSR2^H171A–tube complex
To elucidate the structural basis for the NAD+ hydrolysis by the tube-activated DSR2, we incubated the substrate (1 mM NAD+) with DSR2^H171A–tube complex, which was obtained by co-expression the catalytically inactive DSR2^H171A mutant and tail tube proteins together in the same bacterial cells (Fig. 2d). We then determined the cryo-EM structure of the DSR2^H171A–tube–NAD+ complex at 3.0-Å resolution, in which approximately 60% density was observed, facilitating the accurate tracing of the NAD+ substrates (Supplementary Fig. 4a–c).

Further 3D classification analysis resulted in a map containing almost all parts of DSR2 at a resolution of 3.4 Å (Fig. 5a, b and Supplementary Fig. 4d–f). In addition to the presence of four distinct densities corresponding to NAD+ within the Sir2 domains, we also observed four densities corresponding to the tail tube protein bound to each CTD of tetrameric DSR2 in a DSR2: tube ratio of 4:4 (Fig. 5a, b), which is distinct from the 4:2 ratio observed for the DSR2-tube complex (Fig. 2f), indicating variability in tube binding ratio. We observed large conformational changes in the CTD by superimposing the DSR2–tube complex and the DSR2^H171A–tube-NAD+ complex, likely due to their different tube binding ratios (Supplementary Fig. 4g).

Looking closely at the four densities corresponding to the NAD+ substrate, the densities representing the adenosine and adenosine-ribose moiety of NAD+ were well defined, while we observed a weak density corresponding to the nicotinamide and nicotinamide-ribose groups, suggesting the flexibility of the nicotinamide ribose of NAD+ when it binds to the DSR2–tube complex (Fig. 5c). This observation aligns well with previous reports indicating that the nicotinamide and nicotinamide-ribose groups can adopt various configurations, even within a single crystal lattice, indicating the conformational flexibility of the nicotinamide group[21,22]. NAD+ typically binds to a pocket that can be subdivided into three distinct sites for catalysis: site A, responsible for adenine and adenine-ribose moiety binding; site B, for

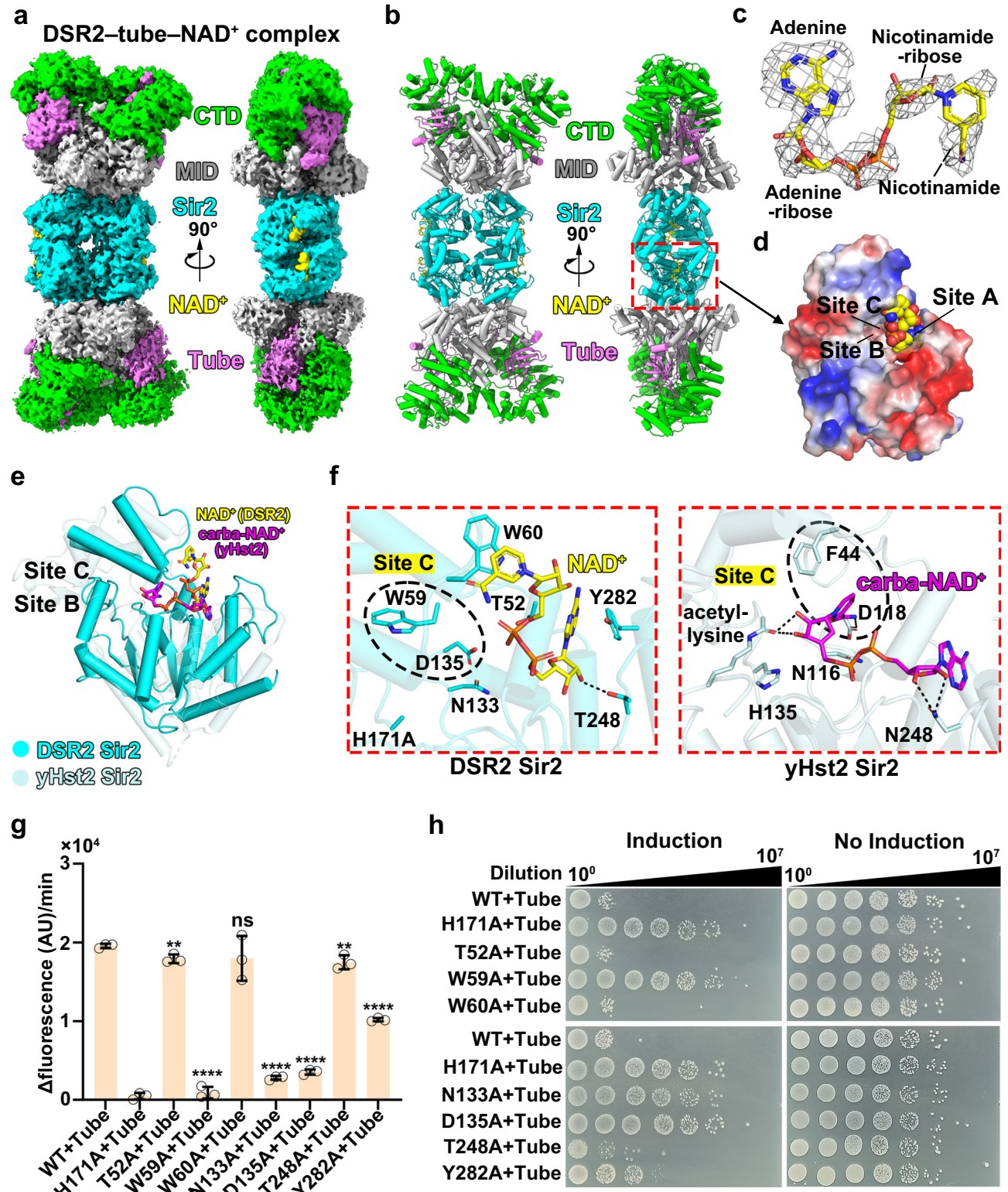

**Fig. 5 | The NAD⁺ substrates bind to the Sir2 domain in the DSR2^H171A–tube–NAD⁺ complex. a, b** Surface (**a**) and ribbon (**b**) representations of the 3.4-Å cryo-EM structure of DSR2^H171A–tube–NAD⁺ complex. **c** Electron density of NAD⁺, shown as gray mesh, contoured at 6.0 $\sigma$. **d** Charge contour of the NAD⁺ binding pocket of DSR2, highlighting three distinct sites (A, B, C) important for catalysis. **e** Structural comparison of the Sir2 domain from DSR2 and from the yHst2 Sir2 deacetylase (PDB 1SZC). **f** Detailed view of the NAD⁺ binding pocket within the Sir2 domain of DSR2 (left) and the yHst2 Sir2 deacetylase (right). **g** NADase activity of WT DSR2 or DSR2 variants in complex with tube proteins ($n = 3$ independent experiments; mean ± SD). The $p$-values were calculated using a one-sided Student's $t$-test [ns: $p > 0.05$, **: $p < 0.01$, ****: $p < 0.0001$; $p = 0.00504$, 0.00000104, 0.196, 0.0000000848, 0.000000191, 0.00909, 0.000000753, (from left to right)]. **h** Survival status of *E. coli* cells co-producing DSR2 (WT or variants) and phage tail tube proteins simultaneously. Source data are provided as a Source Data file.

nicotinamide-ribose binding; and site C, for nicotinamide moiety binding[1,21](Fig. 5d). However, we did not observe a clear density located in sites B and C; by contrast, bound NAD+ within the Sir2 domain of DSR2 adopts an orientation with a ~ 90° rotation compared to that seen with a nonhydrolyzable NAD+ analogue, carba-NAD+, which sits in a position ready for catalysis in yHst2 in yeast (*Saccharomyces cerevisiae*) (Fig. 5e)[23]. Instead of being inserted into sites B and C, the nicotinamide-ribose and nicotinamide groups interacted with the small helical domain, with the nicotinamide moiety stacking with residue W60 (Fig. 5e). In site A, we observed the stabilization of the adenine group through stacking interactions with residues Y282 and T52, while residue T248 forms a hydrogen-bonding interaction with the 2'-OH group of the adenine-ribose (Fig. 5f, left inset). Mutation of Y282 to alanine decreased DSR2 NADase activity by about 50%, whereas the T52A, T248A and W60A single mutants had a negligible effect on DSR2 NADase activity, highlighting the critical role of the Y282-mediated adenine stacking interaction in site A (Fig. 5g). These findings are further supported by our in vivo studies, in which co-expression of Y282A mutant with the tail tube protein resulted in lower toxicity to bacterial cells, while the co-expression of the T52A, T248A, or W60A single mutants was as toxic to bacterial cell growth as WT DSR2 (Fig. 5h). The conserved indispensable residue H171 (Supplementary Fig. 2f, g), which typically serves as a general base for deprotonating the OH group of the nicotinamide ribose, a crucial step in deacetylation of eukaryotic sirtuins[1,8], is far away from the nicotinamide-ribose group (Fig. 5f). Thus, we speculated that the observed NAD+ bound in the DSR2[H171A]–tube complex represents an inactive conformation. However, the weak density of the nicotinamide-ribose and nicotinamide groups of NAD+ in the current conformation suggests that NAD+ might also adopt an alternative conformation in this tube-activated complex (Fig. 5c).

The nicotinamide group of NAD+ should bind to a conserved nicotinamide binding pocket within site C, known as the C pocket, to initiate NAD+ catalysis, allowing a nucleophilic attack on the nicotinamide-ribose[1,21,24] (Fig. 5d). In previously reported structures of Sir2 proteins, NAD+ adopts the active configuration only in the presence of the acetyl-lysine-containing peptide substrate[1,22,24]. However, the channel responsible for the binding of acetyl-lysine-containing peptide substrates is blocked by the presence of α11 (Supplementary Fig. 2e), indicating that the Sir2 domain within DSR2 is not an NAD+-dependent deacetylase. Nevertheless, a conserved aspartic acid (D135 in DSR2; D118 in yHst2) in the C pocket forms hydrogen-bonding interactions with the nicotinamide group, while a conserved phenylalanine residue (W59 in DSR2; F44 in yHst2) forms a π-stacking interaction with the nicotinamide moiety within the C-pocket (Fig. 5f, Supplementary Fig. 2f, g)[1]. The individual mutation of D135 or W59 into alanine significantly impaired the DSR2 activation both in vitro and in vivo (Fig. 5g, h), highlighting the critical role of the C pocket in NAD+ hydrolysis. Collectively, these findings indicate that the Sir2 domain of DSR2 possesses the conserved NAD+ binding pocket seen in other Sir2 deacetylases, but it functions as a NAD+ hydrolase rather than an NAD+-dependent deacetylase.

### Phage-encoded DSAD1 allosterically prevents tube binding for DSR2 activation

The phage-encoded protein DSAD1 (DSR anti-defense 1) directly binds to and inhibit DSR2 activity[8]. Co-expression of DSAD1 together with the tail tube protein and DSR2 proteins in *E.coli* cells resulted in no toxicity to bacterial cells (Fig. 6a), in agreement with the notion that DSAD1 prevents the tube-mediated DSR2 activation[8]. Previous biochemical assays have indicated that DSAD1 binding prevents the tail tube protein from binding to DSR2, thus inhibiting the tube-mediated DSR2 activation[8]. To elucidate the molecular mechanism underlying DSAD1-mediated DSR2 inhibition, we first purified the DSR2–DSAD1 complex, which showed no NADase activity (Fig. 6b, Supplementary

Fig. 5a), and then determined a cryo-EM structure of the DSR2–DSAD1 complex at a 2.5-Å resolution, in which approximately 60% density was observed, facilitating the accurate tracing of DSAD1 and CTD domains (Supplementary Fig. 5b–5d). Further 3D classification resulted in a map containing almost all parts of DSR2 at a resolution of 2.6 Å (Fig. 6c–e, Supplementary Fig. 5e–h). The overall architecture of the tetrameric DSR2–DSAD1 complex closely resembles that of the apo complex (Fig. 1f). Each DSAD1 binds to the CTD[open] of protomers 2 and 4 (Fig. 6c–e), with the binding pocket of DSAD1 located on the backside of where the tail tubes bind (Fig. 6f). This binding of DSAD1 is distinct from the preference of the tail tube protein for CTD[closed] in protomers 1 and 3 (Fig. 2f), resulting in a DSR2: DSAD1 ratio of 4:2.

In contrast to the preference for binding to CTD[closed] of protomers 1 and 3 (Fig. 3a), superimposing of the DSAD1-bound protomer with either protomer 1 containing CTD[closed], or protomer 2 containing CTD[open] from the apo form of DSR2 reveals more pronounced conformational changes in CTD[closed] of protomer 1 that are needed for DSAD1 binding compared the CTD[open] in protomer 2 (Fig. 6g). This observation suggests that the pocket in CTD[open] in protomers 2 and 4 is better suited for DSAD1 binding than CTD[closed] of protomers 1 and 3. When attempting to dock DSAD1 into CTD[closed] of protomer 1, we observed a notable clash between DSAD1 and CTD[closed] (Supplementary Fig. 5i), supporting the idea that DSAD1 exhibits a preference for binding to CTD[open].

A structural comparison using Dali search revealed that DSAD1 shares no structural similarity to reported protein structures. DSAD1 consists of eight β strands and one α helix between β7 and β8 (Fig. 6h). Notably, in addition to the extensive interaction with CTD[open] of protomers 2, the β1 strand of DSAD1 forms an anti-parallel β-strand interaction with the β strand from the MID domain of protomer 1 (Fig. 6h). This β strand is flexible within the tetrameric DSR2 apo complex (Supplementary Fig. 5j). This anti-parallel β-strand interaction locks the position of protomer 1 and prevents the ~6 Å conformational changes of protomer 1 necessary for tube binding, thereby preventing the tube binding (Fig. 6i). Substitution of the β1 strand of DSAD1 (residues L14–S18) with a proline linker (DSAD1[β1] mutant), which disrupts the β-strand-mediated interaction between DSR2 and DSAD1, resulted in notable toxicity to bacterial cells upon co-expression of DSR2 with the tube and DSAD1[β1] mutant (Supplementary Fig. 5k). This results suggested that DSAD1[β1] mutant exhibited reduced inhibition of DSR2 activation, highlighting the importance of this β-strand-mediated interaction. Taken together, these observations indicate that binding of DSAD1 to CTD[open] in protomers 2 and 4 allosterically prevents the binding of the tube protein to CTD[closed] in protomers 1 and 3, thus preventing the tube-mediated activation of DSR2.

### Conformational changes triggered by tube binding activate DSR2 NADase activity

To further investigate the molecular mechanism of tube-mediated DSR2 activation and DSAD1-mediated DSR2 inhibition, we initially considered whether the binding of DSAD1 might prevents the binding of NAD+ to DSR2, leading to an inhibition of its NADase activity. To test this hypothesis, we incubated the 1 mM NAD+ with the DSR2–DSAD1 complex and determined a 2.9-Å cryo-EM structure of the resulting DSR2–DSAD1–NAD+ complex (Supplementary Fig. 6a–e). We observed a strong density corresponding to the NAD+ molecule (Supplementary Fig. 6f), occupying the same position as NAD+ in the DSR2[H171A]–tube complex (Supplementary Fig. 6g). In contrast to the weak density observed for the nicotinamide and nicotinamide-ribose groups of NAD+ bound to the DSR2[H171A]–tube complex (Fig. 5c), the corresponding density observed in the DSR2–DSAD1–NAD+ complex is much stronger (Supplementary Fig. 6f), indicative of stable NAD+ binding. The NAD+ present in the DSR2–DSAD1–NAD+ complex adopts an inactive conformation, as the conserved catalytic residue H171 is unable to access the nicotinamide-ribose group for catalysis

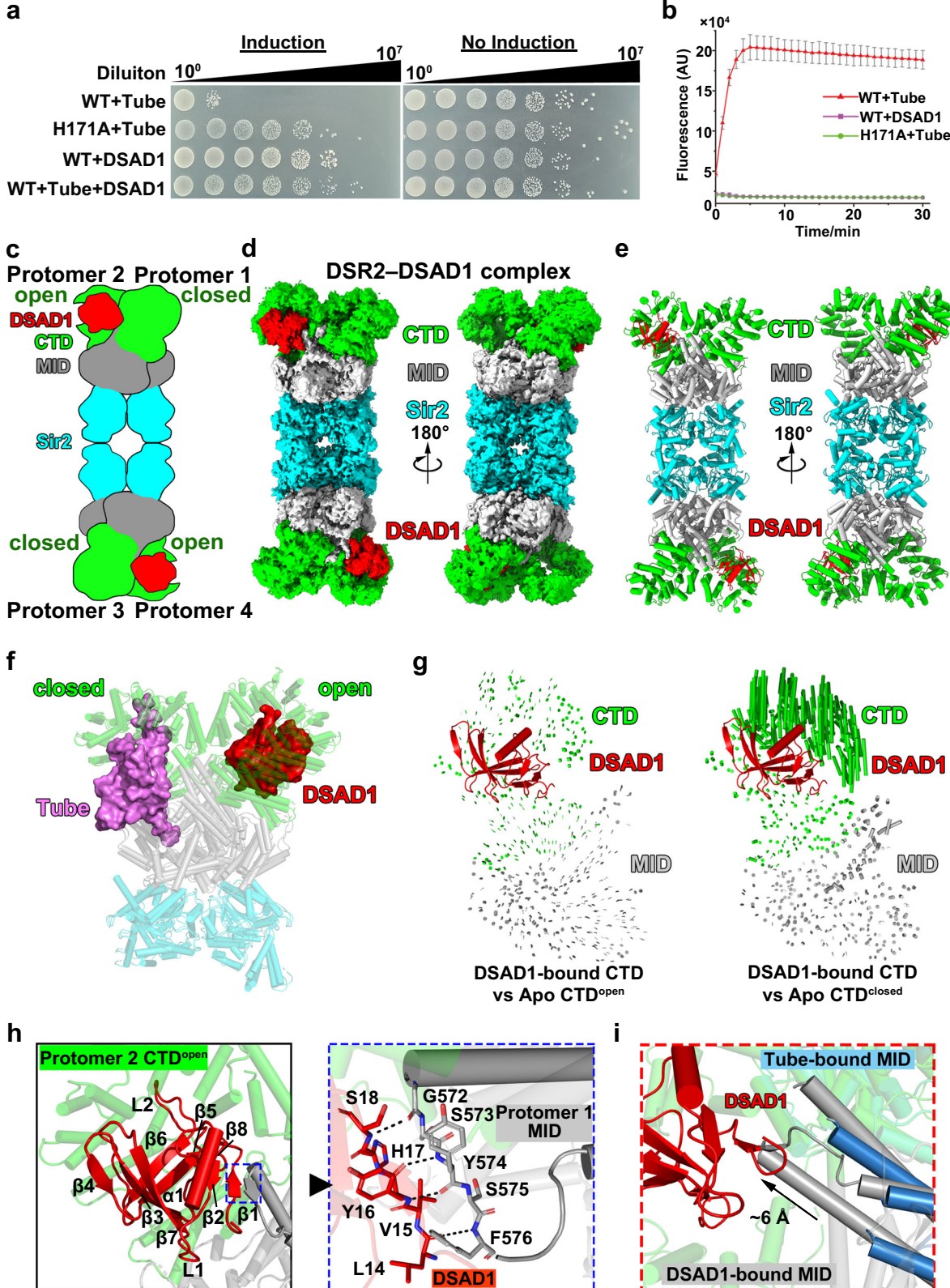

**Fig. 6 | Recognition of phage-encoded anti-DSR2 DSAD1 by CTD^open in DSR2 protomer 2 and 4. a** Survival status of *E. coli* cells co-producing DSR2 and phage tail tube alone or in the presence of DSAD1. **b** NADase activity of WT DSR2, DSR2–DSAD1 and DSR2^H171A–tube complexes (n = 3 independent experiments; mean ± SD). **c–e** Diagram (**c**), surface (**d**) and ribbon (**e**) representations of the 2.6-Å cryo-EM structure of the DSR2–DSAD1 binary complex. **f** Structural comparison between tube-bound and DSAD1-bound of DSR2 protomer1 and protomer

2 showing that the tube protein preferentially binds to CTD^closed, while DSAD1 prefers to bind to CTD^open. **g** Structural comparison of DSAD1-bound CTD with CTD^open (left) and CTD^closed (right) of the apo form of DSR2. Vector length correlates with the scale of domain movement. **h** Interaction interface between DSAD1 and DSR2. The inset to the right shows the detailed interaction between DSAD1 and MID in protomer 1. **i** Structural comparison between DSAD1-bound MID and tube-bound MID. Source data are provided as a Source Data file.

(Supplementary Fig. 6h). Superimposing the structures of the DSR2–DSAD1 and NAD+-bound DSR2–DSAD1 complexes revealed minimal conformational changes in DSR2 upon NAD+ binding (Supplementary Fig. 6i). These findings indicate that the DSR2–DSAD1 complex can bind to NAD+ but does not catalyze its hydrolysis.

We then investigated the conformational changes in DSR2 triggered by tail tube or DSAD1 binding. Superimposing the structures of apo DSR2 with either DSAD1-bound or tube-bound DSR2 revealed minimal conformational changes upon DSAD1 binding (Fig. 7a) with an RMSD of 0.425 Å, but significant conformational changes in the CTD and MID domains upon phage tail tube binding with an RMSD of 2.419 Å (Fig. 7b). To investigate the transfer of tube-binding signal to

the Sir2 domain, we focused on the interface between the Sir2 domain and MID. We replaced three regions located in the interface between the Sir2 domain and MID with GS linkers: loop 1 (residues W143–Y148) within the Sir2 domain; loop 2 (T520–F522) within the adaptor MID; and loop 3 (T297–T303) linking Sir2 and MID domain (referred to as loop1 mutant, loop2 mutant, and loop3 mutant, respectively) (Supplementary Fig. 6j). Co-expression of the loop1 mutant with the phage tail tube was not toxic to *E. coli* cells, whereas the loop2 and loop3 mutants was as toxic to cells as WT DSR2 (Fig. 7c), indicating that the loop 1 within the Sir2 domain plays a crucial role in activating DSR2 NADase activity. To further confirm the importance of loop 1, we purified the DSR2loop1 mutant–tube complex and tested its NADase

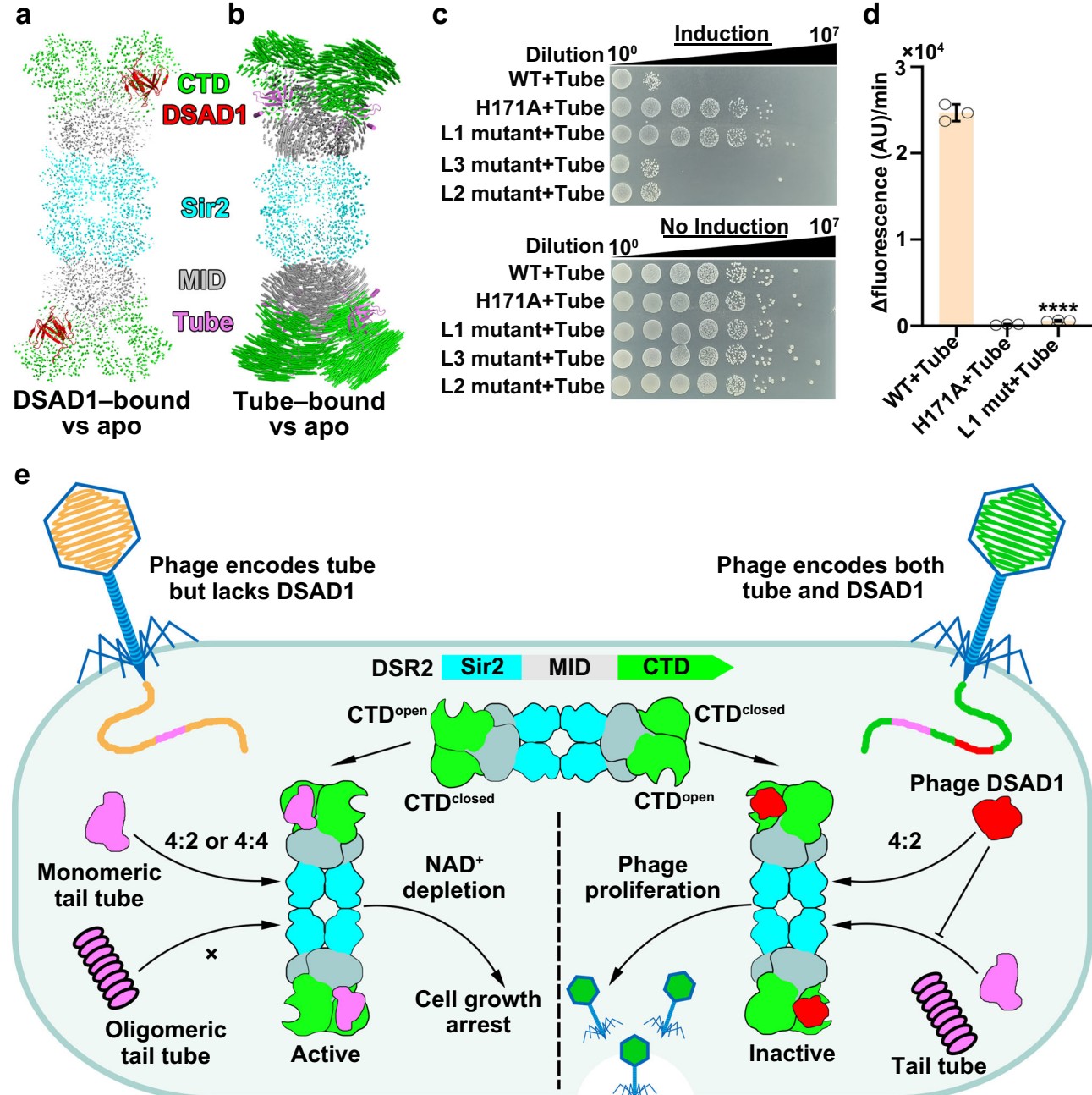

**Fig. 7 | Tube binding–induced conformational changes activate the NADase activity of DSR2. a, b** Structural comparison between the apo form of DSR2 and DSAD1-bound DSR2 (**a**) or tube-bound DSR2 (**b**). **c** Survival status of *E. coli* cells co-producing DSR2 (WT or variants) and tube proteins. **d** NADase activity of WT DSR2 and DSR2–tube variant complexes (*n* = 3 independent experiments; mean ± SD).

The *p*-values were calculated using a one-sided Student's *t*-test [****: *p* < 0.0001; *p* = 0.00000089]. **e** Proposed mechanistic model of DSR2 activation in response to invading phage and the escape of phages from bacterial DSR2 defense system. Source data are provided as a Source Data file.

activity in vitro. Consistent with our in vivo observations, the DSR2[loop1] mutant–tube complex exhibited a near complete abrogation of its NADase activity (Fig. 7d), highlighting the crucial role of this loop within the Sir2 domain for DSR2 NADase activation. Loop1 likely participates in transferring signals from the CTD and MID to the NAD[+] catalytic pocket within the Sir2 domain, thereby activating the NADase activity of the Sir2 domain within the DSR2 tetramer.

## Discussion

Sir2 proteins typically catalyze protein deacetylation using NAD[+] as a cofactor[1]. However, recently identified Sir2 domain–containing proteins participate in bacterial anti-phage defense by directly depleting the intracellular NAD[+] pool[8,10,11,13,14]. In this study, we determined multiple cryo-EM structures of the Sir2 domain–containing protein DSR2, including its inactive DSR2 apo form, the DSR2–DSAD1 binary and the DSR2–DSAD1–NAD[+] ternary complexes, and the active DSR2–tube binary and DSR2–tube–NAD[+] ternary complex. The structural analysis, in combination with in vivo and in vitro biochemical studies, provide mechanistic insights into the bacterial DSR2-mediated anti-phage defense systems.

In our determined tube-bound DSR2 structures, we observed that DSR2 interacts with phage tail tube protein in a ratio of either 4:2 in the DSR2–tube complex or 4:4 in the DSR2[H171A]–tube–NAD[+] complex (Figs. 2g, 5b). We speculated that this variation in binding ratio can be attributed to the different purification methods we used. Given that co-expression of wild-type DSR2 with the tail tube proteins was toxic to bacterial cells (Fig. 1a), we separately expressed wild-type DSR2 and the tube proteins, and then mixed the cell pellets for the subsequent ultrasonication to obtain DSR2–tube complex, while we co-expressed DSR2[H171A] with the tube protein together in the same bacterial cells. Considering the tendency of the tube proteins to oligomerize once translated (Fig. 2a), the resulting lower yield of the monomeric tube proteins might have been insufficient to achieve a 4:4 binding ratio with DSR2. The tendency of tail tube proteins to oligomerize in vitro has also been reported in other phages[19]. This insufficient accumulation of monomeric tube proteins might have led them to first bind to their preferred regions (Fig. 2g). This hypothesis is supported by our biochemical assays, which revealed a significant presence of free apo DSR2 when DSR2 and tube proteins were co-purified (Supplementary Fig. 7a), indicating an insufficient quantity of tube proteins to saturate DSR2 and leading to the preferential binding to specific regions. While we observed a DSR2: tube binding ratio of either 4:2 or 4:4, we were unable to answer whether these different binding ratios influence the strength of the DSR2 NADase activity. Nevertheless, binding of two tube proteins in a DSR2: tube ratio of 4:2 was sufficient for DSR2 activation (Fig. 2d), indicating that binding of tube proteins to DSR2 in a tube: DSR2 ratio of either 2:4 or 4:4 activates the NADase activity of DSR2.

However, we observed a DSR2:DSAD1 binding ratio of 4:2 in both the DSR2–DSAD1 and DSR2–DSAD1–NAD[+] complexes when DSR2 and DSAD1 were co-expressed in the same bacterial cells. Affinity chromatography analysis further confirmed an excess of DSAD1 (Supplementary Fig. 7b), ruling out the possibility that the DSR2:DSAD1 binding ratio of 4:2 was due to the inability of DSAD1 to saturate DSR2, as observed with the tube proteins (Supplementary Fig. 7a). These findings indicate that the binding of DSAD1 to DSR2 in a DSR2:DSAD1 binding ratio of 4:2 is sufficient to inhibit tube-mediated DSR2 activation.

In previously reported Sir2 proteins, catalysis involves the binding of an acetyl-lysine-containing peptide and NAD[+]. The acetyl-lysine group in the peptide substrate contributes to the binding of NAD[+] in the active conformation and is responsible for NAD[+] catalysis. However, as we did not detect NAD[+] bound to the Sir2 domain in the active conformation, we docked the active NAD[+] into the Sir2 domain of DSR2 by superimposing it with that of yHst2 (Supplementary Fig. 7c). The NAD[+] molecule fits well within the catalytic pocket, with the catalytic residues N133 and H171 (corresponding to N116 and H135 in yHst2) essentially superimposed. The N116 residue in yHst2 is involved in initiating the nucleophilic attack at the C1′ atom of the ribose ring for nicotinamide cleavage, and this attack is mediated by an ordered water molecule instead of acetyl-lysine, given that the acetyl-lysine is positioned 4.8 Å away from the C1′ atom of the ribose[23]. This organization might also be the case for the Sir2 domain of DSR2, given that no acetyl-lysine substrates are needed for the NAD[+] catalysis and that mutation of N133 significantly impaired the NADase activity of the Sir2 domain, indicating the critical role of N133 in NAD[+] catalysis (Fig. 2d). Additionally, the conformational changes of the bound NAD[+] substrate induced by the activator's binding could also contribute to the NAD[+] hydrolysis, as observed in the activation of the Csm6 effector of CRISPR type III-A system[25].

We proposed a mechanistic model for both tail tube-mediated DSR2 activation and DSAD1-mediated inhibition. DSR2 exists as head-to-head inactive tetramers within bacterial cells, which is distinct from the NAD[+]-consuming, Toll/interleukin-1 receptor (TIR) domain-containing proteins that depend on the oligomerization of the TIR domain for NADase activation[26]. DSR2 comprises an N-terminal Sir2 effector, a C-terminal CTD sensor and the middle MID adaptor. CTD adopts a closed conformation in protomers 1 and 3, while open in protomers 2 and 4. Upon recognition of the monomeric phage tail tube protein by CTD[closed] in protomers 1 and 3, the CTD undergoes significant conformational changes, which transmit signals to the Sir2 catalytic domain through the interface between the MID and the Sir2 domains (Fig. 7b, e and Supplementary Fig. 6j). This conformational change might contribute to the release of the nicotinamide moiety of NAD[+] from the inactive conformation, but insertion of the nicotinamide moiety of NAD[+] into the conserved C pocket adopts a destabilized active conformation for NAD[+] hydrolysis (Fig. 5e). The depletion of NAD[+] triggers the growth arrest or cell death of infected bacterial cells, thereby inhibiting phage propagation and protecting the whole bacterial population. In response, phages have evolved an anti-DSR2 protein to counter the bacterial DSR2-mediated defense systems. The phage-encoded protein DSAD1 binds to the CTD[open] in protomers 2 and 4, allosterically preventing tube binding to CTD[closed] in protomers 1 and 3 (Fig. 6h), thereby inhibiting the tube-mediated activation of the DSR2 anti-phage defense systems to allow phage propagation (Fig. 7e).

## Methods

### Bacterial strains

*Escherichia coli* DH5α and BL21 Star (DE3) cells were used for plasmid reconstruction and protein expression, respectively.

### Plasmid cloning, protein expression and purification

The codon-optimized genes *dsr2* and *dsad1* were synthesized by Sangon Biotech and cloned into pRSFDuet. This plasmid carried an N-terminal 6×His-tag on *dsr2* at MCS1 and a C-terminal StrepII tag on *dsad1* at MCS2. The codon-optimized gene SPR_tube was also synthesized by Sangon Biotech and cloned into pGEX-6P-1 with an N-terminal GST-tag. Plasmids were transformed into Escherichia coli BL21 Star (DE3) and cultivated at 220 rpm and 37 °C. Protein expression was induced by adding 0.4 mM isopropyl-β-D-thiogalactoside (IPTG, Sangon) when the OD_{600} reached ~0.8. After incubation for 20 h at 18 °C, cells were harvested and resuspended in the lysis buffer (20 mM Tris-HCl pH 7.5, 500 mM NaCl, 1 mM DTT, 20 mM imidazole), and stored at −80 °C.

For the purification of DSR2, harvested cells were resuspended in the lysis buffer and lysed by AH-1500 High Pressure Homogeniser (ATS, inc.) at 800 bar for 10 min. Cell debris was removed by centrifugation at 32,914 g for 30 min. The supernatant was loaded onto a Ni-NTA resin (Genescript), washed with the lysis buffer and the target proteins were eluted using elution buffer (20 mM Tris-HCl pH 7.5, 100 mM NaCl, 400 mM imidazole, 1 mM DTT). The eluate was

subsequently dialyzed several times against a low-salt buffer (20 mM Tris-HCl pH 7.5, 100 mM NaCl) using Amicon Ultra Centrifugal Filters (Millipore Sigma) at 5000 g for 20 min at 8 °C to remove the imidazole and then loaded into a 5 mL HiTrap Q Fast Flow column (Cytiva). The proteins were eluted by using a linear gradient from 100 mM to 1 M NaCl in 20 column volumes, and further purified on a Superdex 200 Increase 10/300 GL column (Cytiva) in a buffer containing 20 mM Tris-HCl, pH 7.5, 150 mM NaCl, and 1 mM DTT.

For the purification of DSR2-DSAD1, the process from cell lysis to passing through the nickel column was the same as mentioned above. The eluate from Ni-NTA was subjected to further purification by loading it into a 5 mL StrepTrap HP column (Smart Lifesciences). Proteins were eluted in buffer A (20 mM Tris-HCl, 100 mM NaCl, 1 mM DTT, pH 7.5) supplemented with 20 mM D-biotin and then loaded into a 5 mL HiTrap Q Fast Flow column (Cytiva). Further purification was carried out on a Superdex 200 Increase 10/300 GL column (Cytiva).

For the purification of DSR2-SPR_tube, the bacteria expressing DSR2 were mixed with bacteria expressing SPR_tube, and the mixture underwent the same lysis process as previously described. The process from cell lysis to passing through the nickel column is the same as mentioned above. The eluate from Ni-NTA was subjected to further purification by loading it onto a glutathione resin (Smart Lifesciences). The GST-affinity tag was removed by overnight incubation with buffer A containing HRV 3 C protease at 4 °C. Then proteins were subjected to a 5 mL HiTrap Q Fast Flow column (Cytiva) and further purification was carried out on a Superdex 200 Increase 10/300 GL column (Cytiva).

All mutants were generated by site-directed mutagenesis and purified using the same method as described above. Primers were provided in the Supplementary Data 1.

### Cryo-EM sample preparation and data acquisition
Aliquots of 3.5 μL DSR2 and DSR2-DSAD1 samples (~25 mg/mL) were applied to the glow-discharged grids (UltrAuFoil 300 mesh R1.2/1.3, Quantifoil), respectively. As for DSR2-tube, the glow discharged grid was pretreated with 0.01% (w/v) polylysine (Sigma Aldrich) to improve the adhesion of proteins to the grid. Then, 3.5 μL DSR2-tube (~4 mg/mL) was applied to the polylysine-treated grids. The grids were blotted for 2.5 s and plunge-frozen in liquid ethane vitrified by liquid nitrogen using Vitrobot Mark IV (FEI Company) at 8 °C and 100% humidity. The grids were transferred to FEI Titan Krios electron microscope operating at 300 kV, and movies (32 frames, total accumulated dose 50 e⁻/Å²) were collected using a direct electron detector Gatan K3 in the counting mode with a defocus range from −1.5 to −2.5 μm. Automated single-particle data acquisition was performed with the EPU (Thermo Fisher Scientific) program at a nominal magnification of 105,000, yielding a final pixel size of 0.827 Å. All other cryo-EM samples were prepared and data acquired using the same methods as mentioned above.

### Cryo-EM data processing
Image processing was performed by cryoSPARC v3.1[27]. A total of 1,631,393 particles were auto-picked using cryoSPARC v3.1[27]. After several iterations of 2D classifications, particles from the best classes were selected and subjected to 3D reconstruction with the initial model generated by cryoSPARC v3.1[27] as a reference. Particles corresponding to the best class were further selected and subjected to non-uniform refinement in cryoSPARC v3.1[27]. All cryo-EM reconstructions were estimated with the gold standard Fourier shell correlation using the 0.143 threshold[28]. Local resolution estimates were calculated from two half data maps in cryoSPARC v3.1[27]. The details related to data processing were shown in Table S1.

### Model building and refinement
With the assistance of bulky residues and PSIPRED secondary structure prediction[29], along with structural models predicted by Robetta

(https://robetta.bakerlab.org/), we manually built the atomic models interactively in COOT[30]. Real-space refinement in PHENIX[31] was used to refine all models against the cryo-EM maps by applying geometric and secondary structure restraints. All structure figures were prepared in PyMol (http://www.pymol.org) and ChimeraX[32]. Low resolution parts of the DSR2 apo complex, DSR2-DSAD1 complex, DSR2-tube complex and DSR2-tube-NAD⁺ complex were rigid-body fit from their corresponding high-resolution models in Chimera[33].

### In vitro NADase activity assay
In the fluorescence assay, 1, N⁶-etheno-adenine dinucleotide (εNAD⁺, Sigma Aldrich, N2630), the fluorescent analogue of NAD⁺, was used as the substrate. Enzymatic activity was characterized by monitoring the increase in fluorescence resulting from cleavage of the quenching nicotinamide group in 1, N⁶-ethenoadenine diphosphate ribose. εNAD⁺ stocks were made using a buffer (100 mM Tris 7.5). The reaction buffer was supplemented with 10 mM MES pH 6.5, 75 mM KCl, 2 mM MgCl₂. The concentrations of proteins and the substrate were optimized to be 400 nM and 50 μM, respectively.

The reaction procedure was carried out in triplicate at 37 °C in a 96-well half area plate (Corning, 3690). Fluorescence intensity was measured by Agilent BioTek Synergy H1, with an excitation wavelength of 300 nm and an emission wavelength of 410 nm, with readings taken every minute over a 0.5 h-1 h duration. The change in fluorescence over time was determined by calculating the slopes of the linear component of the curves. Data analysis was performed using GraphPad Prism 9.

### In vivo growth curve assays
Plasmids pRSFDuet-1-DSR2 and pETDuet-SPR_tube were co-transformed into E. coli BL21-AI competent cells. These cells were cultured in the LB media supplied with 50 μg/mL ampicillin and 50 μg/mL kanamycin. After overnight incubation, the bacterial suspensions were diluted 1:100 in LB media. Cells were grown at 37 °C until the OD₆₀₀ reached 0.3, then the cells were diluted 1:1000 in LB medium supplied with or without inducers (1% L-arabinose and 0.5 mM IPTG), while 2% glucose was added in non-induced medium to inhibit the protein expression. Cells were dispensed (150 μL) into a 96-well plate. And optical density measurements at a wavelength of 600 nm were taken every 10 min using a BioTek Synergy H1 at 37 °C.

### In vivo growth toxicity assays
Plasmids pRSFDuet-1-DSR2 (wild-type or mutant variants) and pETDuet-SPR_tube (wild-type or mutant variants) were co-transformed into E. coli BL21-AI strain cells. These transformed E. coli cells were cultured overnight in LB media, and plasmids were maintained by selection with 50 μg/mL ampicillin and 50 μg/mL kanamycin. The following day, 4 mL LB media was used to dilute the bacterial solution by a factor of 1/100, supplemented with the appropriate antibiotics. When the cells were grown at 37 °C to an OD₆₀₀ of 0.3, cells were collected and resuspended with fresh LB without glucose. Resuspended cells were serially diluted in 10-fold gradient and 10 μL aliquots were then taken from each dilution and pipetted onto LB agar plates supplemented with the additives (2% glucose for inhibiting proteins expression or 1% L-arabinose and 0.5 mM IPTG for induced expression) and the appropriate antibiotics. The plates were subsequently incubated at 37 °C overnight.

### Mass photometry
Mass spectrometry experiments were performed using a Refeyn OneMP (Refeyn Ltd.) at room temperature. Samples were evaluated with microscope rectangular 24 × 50 mm coverslips. The coverslips were cleaned by sonication for 20 min in isopropanol followed by another 20 min in ddH₂O, and then dried using clean nitrogen gas. A plastic gasket (CultureWell TM) was placed on top of the coverslip to create reaction chambers. One drop of silicon oil was applied to the

lens, and a sample of 10 μL was pipetted into the reaction chambers. Image analysis was performed with the software provided by Refeyn Ltd., with the default settings provided by the manufacturer. Raw MP data were processed in DiscoverMP software (Refeyn, Oxford, UK) and plotted as molar mass distribution histograms.

## Reporting summary

Further information on research design is available in the Nature Portfolio Reporting Summary linked to this article.

## Data availability

The cryo-EM density maps have been deposited in the Electron Microscopy Data Bank (EMDB) under accession number EMD-37919 (DSR2 apo complex); EMD-37920 (DSR2 apo (partial) complex); EMD-37921 (DSR2-tube complex); EMD-37922 (DSR2$^{H171A}$-tube-NAD$^+$ complex); EMD-37923 (DSR2$^{H171A}$-tube-NAD$^+$ (partial) complex); EMD-37924 (DSR2-DSAD1 complex); EMD-37925 (DSR2-DSAD1 (partial) complex) and EMD-37926 (DSR2-DSAD1-NAD$^+$ (partial) complex). The atomic coordinates have been deposited in the Protein Data Bank (PDB) with accession number 8WY8 (DSR2 apo complex); 8WY9 (DSR2 apo (partial) complex); 8WYA (DSR2-tube complex); 8WYB (DSR2$^{H171A}$-tube-NAD$^+$ complex); 8WYC (DSR2$^{H171A}$-tube-NAD$^+$ (partial) complex); 8WYD (DSR2-DSAD1 complex); 8WYE (DSR2-DSAD1 (partial) complex) and 8WYF (DSR2-DSAD1-NAD$^+$ (partial) complex).This paper does not report original code. Source data are provided with this paper. Any additional information required to reanalyze the data reported in this paper is available from the lead contact upon request. Source data are provided with this paper.

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

## Acknowledgements

This work was supported by the National Natural Science Foundation of China (grant no. 32270050 to N.J.), the Guangdong and Shenzhen Natural Science Foundation (grant no. 2023A1515012420 and JCYJ20220530114409022 to N.J.), the Guangdong Provincial Science and Technology Innovation Council Grant (2017B030301018), Natural Science Foundation of Guangdong Province, China (Grant No. 2022A1515012143). N. J. is an investigator of SUSTech Institute for Biological Electron Microscopy. We thank the staff at Southern University of Science and Technology (SUSTech) Cryo-EM Center for assistance in data collection on the SUSTech Titan KRIOS cryo-electron microscope.

## Author contributions

N.J. directed the research. X.-Y.L., X.-Y.W. X.-Y.S. and N.C. undertook biochemical studies on sample preparation and purification, biochemical assays. J.-T.Z. performed cryo-EM data collection, data processing, structure refinement and data analysis. Z.L. performed all the in vivo studies. J. Z. and H.L. contribute to helpful discussions. N.J. and J.-T.Z. wrote the manuscript with input from other authors.

## Competing interests

The authors declare no competing interests.
