## [Peer Review File · Nature Communications]

Structural basis for phage-mediated activation and repression of bacterial DSR2 anti-phage defense systemREVIEWER COMMENTS

Reviewer #1 (Remarks to the Author):

The manuscript by Zhang et. al. reported cryo-EM structures of DSR2 in an apo state, an active state, and an inhibited state and revealed mechanisms of DSR2 assembly and regulation. DSR2 is a newly identified anti-phage system in bacteria. DSR2 can be activated by phage tube proteins and can be inhibited by another phage protein called DSAD1. In this manuscript, the authors showed that DSR2 consists of three domains, SIR2, MID, and CTD, and assembles as a tetramer. Further structural analysis revealed that the CTD of DSR2 binds to the tube proteins with a stoichiometry of 4:4 or 4:2, which triggers conformational changes of DSR2 for activating the SIR2 domain. The active DSR2 can hydrolyze NAD⁺ to induce cell death. In addition, the authors also determined cryo-EM structures of the DSR2-DSAD1 to reveal how DSAD1 binding blocks the recruitment of phage tube proteins. Structural guided biochemical analysis and cellular assays further solidified conclusions derived from structural analysis. Together, the findings in this manuscript are endorsed by strong evidence and represent a breakthrough in the field. Minor revision is required before being published.

Minor points:

1. The author did not elaborate how the DSR2H171A-tube-NAD⁺ complex was prepared in the manuscript. Why in this mutant protein the author could see 4 copies of the tube molecule but only see two copies of the tube protein in the wild type DSR2?
2. Can you prepare the complex of wild type DSR2 with 4 copies of the tube protein? Whether the NADase activities are different between DSR2-Tube (4:2) and DSR2-Tube (4:4)?
3. The cryo-EM map in Supplementary Fig. 4f seems to be overfitted with lots of noisy densities.
4. There are a few typos in the manuscript to be corrected.

Line 249: "which is distinct from the 4:2 ratio observed for the in DSR2-tube complex" "in" should be deleted.

Line 261: “gourp” should be “group”.

In the last section of the results: All the “NAD+” should be replaced by “NAD⁺”

Reviewer #2 (Remarks to the Author):

Recently, there has been an explosive renewed interest in the defense strategies employed by bacteria and archaea to defend themselves against phage. The DSR2 antiphage defense system employs a SIR2-domain containing protein to rapidly degrade the critical metabolite NAD⁺ in response to viral infection. While it has previously been shown that the phage tail tube is the viral factor responsible for triggering DSR2 activation, the underlying molecular mechanism is unknown. Phages are known to encode anti-defense mechanisms that allow the virus to overcome or evade the host immune response and complete the replicative cycle to infect other host cells. The DSAD1 phage anti-defense protein was shown previously to counteract DSR2 signaling however the details of this activity are poorly defined.

In this manuscript, Zhang, Liu, and Li et al. provide convincing structural and biochemical evidence for the direct binding of phage tail tube protein and the DSAD1 anti-defense protein to DSR2. Their cryo-EM structures give us a three-dimensional roadmap to understand how oligomerization and conformational changes triggered by viral cue recognition lead to NADase catalytic activity. Additionally, the structures and mutational analysis delineate a probable mechanism for how DSAD1 can prevent these conformational changes and limit NADase ability. Overall, the manuscript is well organized, the experimental design is well thought out and clearly worded, and the figures are aesthetically pleasing. The results are presented in a straightforward and easy to follow way. This manuscript follows up on prior work but provides ample amounts of novel data which are formulated into a compelling story.

I believe this manuscript will be very well received by the scientific community. The results reported here are timely and relevant to the field of antiphage defense and I find that the data generally support the proposed model and conclusions (Fig. 7). Outlined below are suggestions and comments that may aid in productive revision of the manuscript before resubmission for peer-review. In my opinion, additional mutational studies and NADase assays would make for a more complete study. My comments and suggestions are generally minor given the excellent data presentation that the authors provided.

Comments and suggestions:

- Can the authors provide a rationale for why mixing of cell pellets was the way forward rather than mixing of individually expressed and purified protein (DSR2 and tail protein)?
- Did the Q34–K57, Δ Loop1 mutant and the Δ Loop2 mutant of the tail tube protein still interact and form stable complexes with DSR2? Clearly Δ Loop2 still activates DSR2 (Fig. 3e) which suggests the answer will be ‘yes’ but it is less clear for Δ Loop1. Can these data be shown?
- Nomenclature is inconsistent between main text and figures for describing the loop deletion mutants of the tail tube protein. In text, tail tube loop mutants are referred to incorrectly as DSR2 Δ Loop1 or DSR2 Δ Loop2 (see quoted text below) and in Fig. 3 as ‘tube(Δ L1)’. “Co-expression of DSR2 Δ Loop1 with the tail tube protein did not exhibit notable toxicity to bacterial cells, whereas co-expression of DSR2 Δ Loop2 with the tail tube protein remained cytotoxic (Fig. 3e). To verify our *in vivo* observations, we purified the DSR2 Δ Loop1–tube complex and assessed its NADase activity. In agreement with our results above, the DSR2 Δ Loop1–tube complex showed a significant drop in its NADase activity relative to the DSR2–tube complex (Fig. 3f), demonstrating the essential role of loop1 for DSR2 activation.”
- It is unclear why NADase activity data are not reported for the Δ loop2 mutant in Fig. 3f.
- In Fig. 3d, why are there also not any cutaway zoomed/magnified views of the L1 and L2 regions and what they interact with on DSR2? As the mutations/deletions were made of these regions it seems prudent to provide some high-level detail in a figure.
- In Fig. 3e, Δ L2 actually looks like it has a more severe phenotype than WT + tail tube (by \sim 1 fold dilution). Can this be speculated upon in the discussion or explored in more detail? Similarly, in Figure 4b it looks as if N202A,T206A + tube is more active than WT as is the Q610A, R613A mutant.
- Fig. 3e and f labeling should likely be switched based on their physical location in the figure.
- It might be informative to also test the single point mutant variants of the tetramerization interaction residues Y71 and D188. The double mutant clearly has a phenotype but it would also be useful to know the contributions of the individual residues and to know if there is some amount of synergy when mutated in combination.
- In Fig. 4c it might be useful to show NADase data for all mutants tested in panel b.
- It might be informational to test mutagenesis to see if any contacts are critical for mediating DSAD1-DSR2 interactions. A specific beta-strand of DSAD1 is highlighted as making extensive contacts with DSR2 (Fig. 6h) however the importance or relevance of observed DSAD1-DSR2 contacts on DSR2 NADase inhibition are not explored.
- In Fig. 7d it isn’t clear why the L2 + Tube and L3 + Tube NADase activity data are not shown.
- Remove the ‘s’ from hydrolases. “Thus, our findings suggest that the tetrameric DSR2 should function as a tube-activated NAD⁺ hydrolases rather than an NAD⁺-dependent deacetylase.”
- Add ‘2’ after DSR in DSR:tail tube. “To gain structural insights into how the monomeric rather than the oligomeric phage tail tube proteins activate the NADase activity of DSR2, we determined the cryo-EM

structure of the DSR2–tube complex at 3.6-Å resolution (Fig. 2e–g, Supplementary Fig. 3a–c), with a DSR: tail tube ratio of 4:2.”

- Typo- ‘gourp’ should be ‘group’. “This observation aligns well with previous reports indicating that the nicotinamide and nicotinamide-ribose groups can adopt various configurations, even within a single crystal lattice, indicating the conformational flexibility of the nicotinamide gourp”
- Typo in Fig. 7- “oligomeric tail thbe” should be “tube”
- There needs to be a methods section added describing how the experiment and growth curve analysis presented in Fig. 1b was conducted including statistical analysis (are the error bars indicative of technical replicates?).
- References # 8 and # 16 are the same.
- References # 19 and # 22 are the same.
- When discussing Thois antiphage defense, reference should be added to Leavitt et al. (2022) Nature- “Viruses inhibit TIR gcADPR signalling to overcome bacterial defence” which shows that ThsB of Thois produces a 1’-3’gcADPR isomer which binds to the SLOG domain of ThsA and initiates SIR2 domain NADase activity. This work built upon the currently referenced manuscript by Ofir et al. (2021) Nature which originally posits a signaling molecule but did not confirm the identity.
- Remove the ‘s’ from hydrolases. “Thus, our findings suggest that the tetrameric DSR2 should function as a tube-activated NAD⁺ hydrolases rather than an NAD⁺-dependent deacetylase.”

REVIEWER COMMENTS

Reviewer #1 (Remarks to the Author):

The manuscript by Zhang et. al. reported cryo-EM structures of DSR2 in an apo state, an active state, and an inhibited state and revealed mechanisms of DSR2 assembly and regulation. DSR2 is a newly identified anti-phage system in bacteria. DSR2 can be activated by phage tube proteins and can be inhibited by another phage protein called DSAD1. In this manuscript, the authors showed that DSR2 consists of three domains, SIR2, MID, and CTD, and assembles as a tetramer. Further structural analysis revealed that the CTD of DSR2 binds to the tube proteins with a stoichiometry of 4:4 or 4:2, which triggers conformational changes of DSR2 for activating the SIR2 domain. The active DSR2 can hydrolyze NAD⁺ to induce cell death. In addition, the authors also determined cryo-EM structures of the DSR2-DSAD1 to reveal how DSAD1 binding blocks the recruitment of phage tube proteins. Structural guided biochemical analysis and cellular assays further solidified conclusions derived from structural analysis. Together, the findings in this manuscript are endorsed by strong evidence and represent a breakthrough in the field. Minor revision is required before being published.

We greatly appreciate the reviewer for the high evaluation of our work as well as the constructive feedback. We have addressed all the comments listed below and revised the manuscript accordingly. We believe that these changes have substantially enhanced the quality of our manuscript.

Minor points:

1. The author did not elaborate how the DSR2^{H171A}-tube-NAD⁺ complex was prepared in the manuscript. Why in this mutant protein the author could see 4 copies of the tube molecule but only see two copies of the tube protein in the wild type DSR2?

We appreciate the reviewer's insightful comments. In our discussion section (line 414 to 430), we have discussed the variation in binding ratio of tail tube to DSR2 in the DSR2-tube and DSR2^{H171A}-tube-NAD⁺ complex. We proposed that this variation in binding ratio may be attributed to the different purification methods we used.

To enhance the clarity, we have incorporated details on the preparation procedures of both the DSR2-tube (line 162 to 165) and DSR2^{H171A}-tube-NAD⁺ complexes (line 245 to 248) directly in the main text, as the reviewer suggested.

Line 414 to 430

“In our determined tube-bound DSR2 structures, we observed that DSR2 interacts with phage tail tube protein in a ratio of either 4:2 in the DSR2-tube complex or 4:4 in the DSR2^{H171A}-tube-NAD⁺ complex (Fig. 2g, 5b). We speculated that this variation in binding ratio can be attributed to the different purification methods we used. Given that

co-expression of wild-type DSR2 with the tail tube proteins was toxic to bacterial cells (Fig. 1a), we separately expressed wild-type DSR2 and the tube proteins, and then mixed the cell pellets for the subsequent ultrasonication to obtain DSR2–tube complex, while we co-expressed DSR2^{H171A} with the tube protein together in the same bacterial cells. Considering the tendency of the tube proteins to oligomerize once translated (Fig. 2a), the resulting lower yield of the monomeric tube proteins might have been insufficient to achieve a 4:4 binding ratio with DSR2. The tendency of tail tube proteins to oligomerize *in vitro* has also been reported in other phages¹⁹. This insufficient accumulation of monomeric tube proteins might have led them to first bind to their preferred regions (Fig. 2g). This hypothesis is supported by our biochemical assays, which revealed a significant presence of free apo DSR2 when DSR2 and tube proteins were co-purified (Supplementary Fig. 7a), indicating an insufficient quantity of tube proteins to saturate DSR2 and leading to the preferential binding to specific regions.”

Line 162 to 165:

“we first purified the DSR2-tube complex by separate expression of DSR2 and tube proteins, followed by mixing their cell pellets for the subsequent ultrasonication, as coexpression of DSR2 and tail tube proteins are toxic to the bacterial cells.”

Line 245 to 248:

“To elucidate the structural basis for the NAD⁺ hydrolysis by the tube-activated DSR2, we incubated the substrate (1 mM NAD⁺) with DSR2^{H171A}–tube complex, which was obtained by co-expression the catalytically inactive DSR2^{H171A} mutant and tail tube proteins together in the same bacterial cells (Fig. 2d).”

2. Can you prepare the complex of wild type DSR2 with 4 copies of the tube protein? Whether the NADase activities are different between DSR2-Tube (4:2) and DSR2-Tube (4:4)?

We appreciate the reviewer’s constructive comment. The reviewer raised an intriguing point. As mentioned in our response to comment 1, given that coexpression of DSR2 and tube protein is toxic to bacterial cells, the toxicity observed in bacterial cells during coexpression of DSR2 and tail tube proteins enable us to separate expression of these proteins to obtain DSR2-tube complex. However, the yield of monomeric tube protein is very low, with a tendency to oligomerize post-purification. Thus, obtaining sufficient monomeric tube for the DSR2-Tube (4:4) construction was challenging. Additionally, the separation of DSR2-Tube (4:4) and DSR2-Tube (4:2) for comparative activity assessments will also be challenging. Nevertheless, we have demonstrated that the DSR2-Tube (4:2) is active. We appreciate the reviewer’s insightful comments, we have incorporated this limitation into our discussion session (line 431 to 433).

Line 431 to 433:

“While we observed a DSR2: tube binding ratio of either 4:2 or 4:4, we were unable to

answer whether these different binding ratios influence the strength of the DSR2 NADase activity.”

3. The cryo-EM map in Supplementary Fig. 4f seems to be overfitted with lots of noisy densities.

We thank the reviewer’s constructive suggestion. We have adjusted the cryo-EM map in Supplementary Fig. 4f with a higher contour level and revised this figure.

Supplementary Fig. 4: Cryo-EM reconstruction of the DSR2–tube–NAD⁺ ternary complex.

f, Final 3D reconstructed map of the DSR2–tube–NAD⁺ ternary complex, colored according to local resolution.

4. There are a few typos in the manuscript to be corrected.

Line 249: “which is distinct from the 4:2 ratio observed for the in DSR2-tube complex” “in” should be deleted.

Line 261: “gourp” should be “group”.

In the last section of the results: All the “NAD+” should be replaced by “NAD⁺”

We thank the reviewer for pointing out these typos. We have corrected these typos in our revised manuscript.

Reviewer #2 (Remarks to the Author):

Recently, there has been an explosive renewed interest in the defense strategies employed by bacteria and archaea to defend themselves against phage. The DSR2 antiphage defense system employs a SIR2-domain containing protein to rapidly degrade the critical metabolite NAD⁺ in response to viral infection. While it has previously been shown that the phage tail tube is the viral factor responsible for triggering DSR2 activation, the underlying molecular mechanism is unknown. Phages are known to encode anti-defense mechanisms that allow the virus to overcome or evade the host immune response and complete the replicative cycle to infect other host

cells. The DSAD1 phage anti-defense protein was shown previously to counteract DSR2 signaling however the details of this activity are poorly defined.

In this manuscript, Zhang, Liu, and Li et al. provide convincing structural and biochemical evidence for the direct binding of phage tail tube protein and the DSAD1 anti-defense protein to DSR2. Their cryo-EM structures give us a three-dimensional roadmap to understand how oligomerization and conformational changes triggered by viral cue recognition lead to NADase catalytic activity. Additionally, the structures and mutational analysis delineate a probable mechanism for how DSAD1 can prevent these conformational changes and limit NADase ability. Overall, the manuscript is well organized, the experimental design is well thought out and clearly worded, and the figures are aesthetically pleasing. The results are presented in a straightforward and easy to follow way. This manuscript follows up on prior work but provides ample amounts of novel data which are formulated into a compelling story.

I believe this manuscript will be very well received by the scientific community. The results reported here are timely and relevant to the field of antiphage defense and I find that the data generally support the proposed model and conclusions (Fig. 7). Outlined below are suggestions and comments that may aid in productive revision of the manuscript before resubmission for peer-review. In my opinion, additional mutational studies and NADase assays would make for a more complete study. My comments and suggestions are generally minor given the excellent data presentation that the authors provided.

We appreciate the reviewer's high evaluation of our work and the constructive suggestions. We have addressed all the comments listed below with additional experiments and revised the manuscript accordingly. We believe that these changes have significantly improved our manuscript.

Comments and suggestions:

- Can the authors provide a rationale for why mixing of cell pellets was the way forward rather than mixing of individually expressed and purified protein (DSR2 and tail protein)?

We thank the reviewer's insightful comments. Indeed, we initially attempted to purify the DSR2 and tail tube proteins separately. However, the yield of individual monomeric tail tube proteins was very low, with a tendency of oligomerization even after purification. Moreover, the tail tube proteins tend to oligomerize with higher protein concentrations. We then tried to mix the cell pellets together. This strategy aimed to maintain the tube protein at a lower concentration, favoring the presence of more monomeric tube protein. As anticipated, this approach allowed us to obtain the stable DSR2-tube complex.

• Did the Q34–K57, Δ Loop1 mutant and the Δ Loop2 mutant of the tail tube protein still interact and form stable complexes with DSR2? Clearly Δ Loop2 still activates DSR2 (Fig. 3e) which suggests the answer will be ‘yes’ but it is less clear for Δ Loop1. Can these data be shown?

We appreciate the reviewer’s constructive comment. Following the reviewer’s suggestions, we purified the DSR2-tube ^{Δ Loop1} complex. The Δ Loop1 mutant of the tail tube still forms a stable complex with DSR2 in a molecular ratio similar to the wild type (as illustrated in the figure below). This observation suggests the loss of the toxicity might be due to the perturbation rather than destruction in the DSR2-tube interface.

We have added the following text in our manuscript (line 200 to 204) and Supplementary Fig. 3g and h.

Line 200 to 204:

“To verify the loss of the toxicity was not due to the poor expression of tube ^{Δ Loop1} mutant, we purified the DSR2–tube ^{Δ Loop1} complex. Our results revealed that tube ^{Δ Loop1} mutant forms a stable complex with DSR2 in a molecular ratio similar to the wild type (Supplementary Fig. 3g, h).”

Supplementary Fig. 3: Cryo-EM reconstruction of the DSR2–tube complex.

g and h, Size exclusion chromatography and SDS-PAGE profiles of the purified DSR2–tube ^{Δ Loop1} (g) and DSR2–tube^{WT} (h) complexes.

• Nomenclature is inconsistent between main text and figures for describing the loop deletion mutants of the tail tube protein. In text, tail tube loop mutants are referred to incorrectly as DSR2 Δ Loop1 or DSR2 Δ Loop2 (see quoted text below) and in Fig. 3 as ‘tube(Δ L1)’. “Co-expression of DSR2 Δ Loop1 with the tail tube protein did not exhibit notable toxicity to bacterial cells, whereas co-expression of DSR2 Δ Loop2 with the tail tube protein remained cytotoxic (Fig. 3e). To verify our in vivo observations, we

purified the DSR2 Δ Loop1–tube complex and assessed its NADase activity. In agreement with our results above, the DSR2 Δ Loop1–tube complex showed a significant drop in its NADase activity relative to the DSR2–tube complex (Fig. 3f), demonstrating the essential role of loop1 for DSR2 activation.”

We thank the reviewer for pointing out the inconsistent of the nomenclature between main text and figure. We have changed the descriptions for the loop deletion mutants of the tail tube protein to tube ^{Δ Loop1} and tube ^{Δ Loop2}, respectively, and revised our main text and figure accordingly.

Line 196 to 207:

“To further highlight the importance of these two loops, we replaced residues in loop1 (residues Q34–K57, tube ^{Δ Loop1}) or residues in loop 2 (residues F204–P216, tube ^{Δ Loop2}) with a GS linker. Co-expression of DSR2 with the tube ^{Δ Loop1} did not exhibit notable toxicity to bacterial cells, whereas co-expression of DSR2 with the tube ^{Δ Loop2} remained cytotoxic (Fig. 3e). To verify the loss of the toxicity was not due to the poor expression of tube ^{Δ Loop1} mutant, we purified the DSR2–tube ^{Δ Loop1} complex. Our results revealed that tube ^{Δ Loop1} mutant forms a stable complex with DSR2 in a molecular ratio similar to the wild type. In agreement with our *in vivo* observations, the DSR2–tube ^{Δ Loop1} complex showed a significant drop in its NADase activity relative to the DSR2–tube complex (Fig. 3f), demonstrating the essential role of loop1 for DSR2 activation.”

- It is unclear why NADase activity data are not reported for the Δ loop2 mutant in Fig. 3f.

We appreciate the insightful comment from the reviewer. To assess the significance of loop1 and loop2 of tail tube proteins in activating the NADase activity of DSR2, we first performed *in vivo* growth toxicity assays to screen for potential mutants that could impact the toxicity of the tail tube proteins. As Tube ^{Δ loop2} exhibited toxicity similar to wild-type tube, indicating its retention of DSR2 activation *in vitro*. Thus, we focused our evaluation on Tube ^{Δ loop1}, which exhibited a loss of toxicity, to eliminate the possibility of poor mutant expression as a cause for the observed decrease in toxicity *in vivo*.

We have incorporated the following text into our manuscript to ensure clarity and better understanding for our readers.

Lines 200 to 204:

“To verify the loss of the toxicity was not due to the poor expression of tube ^{Δ Loop1} mutant, we purified the DSR2–tube ^{Δ Loop1} complex. Our results revealed that tube ^{Δ Loop1} mutant forms a stable complex with DSR2 in a molecular ratio similar to the wild type (Supplementary Fig. 3g, h).”

- In Fig. 3d, why are there also not any cutaway zoomed/magnified views of the L1 and

L2 regions and what they interact with on DSR2? As the mutations/deletions were made of these regions it seems prudent to provide some high-level detail in a figure.

We thank the reviewer for the insightful suggestion. Following the reviewer's suggestion, we have incorporated zoomed views in Supplementary Fig. 3f to illustrate the interactions of Loop1 and Loop2 with DSR2.

The incorporated supplementary Fig. 3f and its corresponding figure legend are provided below:

Supplementary Fig. 3: Cryo-EM reconstruction of the DSR2–tube complex.

f, The Loop1 (L1) and Loop2 (L2) of the tail tube protein interact with CTD^{closed} and MID domain of protomer 1, respectively. Expanded views indicate the detailed interactions mediated by L1 and L2.

- In Fig. 3e, Δ L2 actually looks like it has a more severe phenotype than WT + tail tube (by ~1 fold dilution). Can this be speculated upon in the discussion or explored in more detail? Similarly, in Figure 4b it looks as if N202A, T206A + tube is more active than WT as is the Q610A, R613A mutant.

We thank the reviewer for the insightful comment. The figures presented in our main figure represent one of the three independent replicates conducted. Considering the toxicity observed in the other two replicates, the observation that the mutant is more active than WT should be considered within the experimental error.

The other two replicates are shown below:

Survival status of *E. coli* cells co-producing DSR2 and phage tail tube (WT or tube^{ΔLoop2}) proteins (a), DSR2 (WT, DSR2^{N202A, T206A} or DSR2^{Q610A, R613A}) and phage tail tube proteins (b) simultaneously.

• Fig. 3e and f labeling should likely be switched based on their physical location in the figure.

We thank the reviewer's suggestion, and we have revised the figure accordingly.

Figure 3

Fig. 3: Recognition of the tail tube protein from phage SPR tail tube by CTD^{closed} in protomers 1 and 3

- It might be informative to also test the single point mutant variants of the tetramerization interaction residues Y71 and D188. The double mutant clearly has a phenotype but it would also be useful to know the contributions of the individual residues and to know if there is some amount of synergy when mutated in combination.

We appreciate the reviewer's constructive suggestion. According to the review's suggestion, we constructed DSR2^{Y71} and DSR2^{D188} single mutants to access individual effects of these mutants through *in vivo* growth toxicity assays. The results revealed that either DSR2^{Y71} or DSR2^{D188} single mutant maintained toxicities comparable to WT DSR2 (supplementary Fig. 3i), highlighting that the loss of activity requires the presence of double mutants. Accordingly, we have revised our main text and incorporate these results into supplementary Fig. 3i.

Line 229 to 233:

“However, the DSR2^{Y71A, D188A} double mutant but not DSR2^{Y71A}, DSR2^{D188A} single

mutants significantly impaired DSR2 activation, resulting in no toxicity to bacterial cells when co-expressed with the tail tube protein (Fig. 4b, Supplementary Fig. 3i), highlighting that the loss of activity in DSR2 activation requires the presence of double mutants.”

Supplementary Fig. 3: Cryo-EM reconstruction of the DSR2–tube complex.
i, Survival status of *E. coli* cells co-producing DSR2 (WT or variants) and tube.

- In Fig. 4c it might be useful to show NADase data for all mutants tested in panel b.

We thank the reviewer for the insightful comment. We agree with the reviewer that testing all mutants *in vitro* would be useful. We reasoned that if the mutants (N202A, D188A double mutant and Q610A, R613A double mutant) maintain the activity *in vivo*, they should also be active *in vitro*. We reasoned that the *in vivo* data should be sufficient to support our conclusions. As our response for the previous mentioned tube ^{Δ loop2} mutant, we performed *in vitro* NADase activity tests for the Y71A, D188A mutant to confirm that the loss of the toxicity observed *in vivo* was not due to the poor expression of the Y71A, D188A mutant.

- It might be informational to test mutagenesis to see if any contacts are critical for mediating DSAD1-DSR2 interactions. A specific beta-strand of DSAD1 is highlighted as making extensive contacts with DSR2 (Fig. 6h) however the importance or relevance of observed DSAD1-DSR2 contacts on DSR2 NADase inhibition are not explored.

We appreciate the reviewer’s constructive suggestion. We agree with the reviewer that mutagenesis testing would be helpful. As residues L14 to S18 within the β 1 strand of DSAD1 are involved in an anti-parallel β -strand interaction with the β strand from the MID domain of protomer 1 primarily through main-chain interactions rather than side chain interaction. To disrupt the β 1-strand-mediated interaction, we substituted residues 14-LVYHS-18 with 14-PPPPP-18, creating the DSAD1 ^{β 1} mutant. As expected,

coexpression of phage tail tube proteins with DSR2 and DSAD1^{β1} mutant showed significant toxicity to bacterial cells (Supplementary Fig. 5k), highlighting the crucial role of this β strand in DSR2-DSAD1 interaction. However, we failed to purify the individual DSAD1 and DSAD1^{β1} mutant due to its instability to test the mutation's impact *in vitro* to eliminate concerns about poor expression of the mutant. Nevertheless, the *in vivo* data indicated the importance of the β1-strand-mediated interaction.

Accordingly, we have revised our main text and incorporate these results into supplementary Fig. 5k (Line 352 to 358).

Line 352 to 358:

“Substitution of the β1 strand of DSAD1 (residues L14–S18) with a proline linker (DSAD1^{β1} mutant), which disrupts the β-strand-mediated interaction between DSR2 and DSAD1, resulted in notable toxicity to bacterial cells upon co-expression of DSR2 with the tube and DSAD1^{β1} mutant (Supplementary Fig. 5k). This results suggested that DSAD1^{β1} mutant exhibited reduced inhibition of DSR2 activation, highlighting the importance of this β-strand-mediated interaction.”

Supplementary Fig. 5: Cryo-EM reconstruction of the DSR2–DSAD1 complex.

k, Survival status of *E. coli* cells co-producing DSR2 (WT or mutant) and phage tail tube alone or in the presence of DSAD1 (WT or mutant).

• In Fig. 7d it isn't clear why the L2 + Tube and L3 + Tube NADase activity data are not shown.

We thank the reviewer for the insightful comment. As our response to previous comments, we reasoned that if the L2 and L3 mutants maintain the activity *in vivo*, they should also be active *in vitro*. The *in vivo* data should be sufficient to support our conclusions. We performed *in vitro* NADase activity tests for the L1 mutant to confirm that the loss of the toxicity observed *in vivo* was not due to the poor expression of the L1 mutant.

• Remove the 's' from hydrolases. “Thus, our findings suggest that the tetrameric DSR2 should function as a tube-activated NAD⁺ hydrolases rather than an NAD⁺-dependent deacetylase.”

We thank the reviewer for pointing out this typo. The extra 's' has been removed from hydrolases. Thank you.

- Add '2' after DSR in DSR:tail tube. “To gain structural insights into how the monomeric rather than the oligomeric phage tail tube proteins activate the NADase activity of DSR2, we determined the cryo-EM structure of the DSR2–tube complex at 3.6-Å resolution (Fig. 2e–g, Supplementary Fig. 3a–c), with a DSR: tail tube ratio of 4:2.”

We appreciate the reviewer for pointing out this typo. We have revised “DSR: tail tube” into “DSR2: tail tube”. Thank you.

- Typo- ‘gourp’ should be ‘group’. “This observation aligns well with previous reports indicating that the nicotinamide and nicotinamide-ribose groups can adopt various configurations, even within a single crystal lattice, indicating the conformational flexibility of the nicotinamide gourp”

We thank the reviewer for pointing out this typo. We have corrected “gourp” to “group”. Thank you.

- Typo in Fig. 7- “oligomeric tail thbe” should be “tube”

We appreciate the reviewer’s thorough reading for our manuscript. We have revised the figure by replacing “thbe” with “tube” in Fig. 7. Thank you very much!

- There needs to be a methods section added describing how the experiment and growth curve analysis presented in Fig. 1b was conducted including statistical analysis (are the error bars indicative of technical replicates?).

We thank the reviewer for the insight comment, which is invaluable in enhancing the accuracy and comprehensiveness of our manuscript.

We have added the related text in the methods section of our manuscript (lines 589 to 598), and revised Fig. 1b represent three independent experiments. We have showed each replicate instead of mean \pm s.d. in Fig. 1b.

***In vivo* growth curve assays**

Plasmids pRSFDuet-1-DSR2 and pETDuet-SPR_tube were co-transformed into *E. coli* BL21-AI competent cells. These cells were cultured in the LB media supplied with 50 $\mu\text{g}/\text{mL}$ ampicillin and 50 $\mu\text{g}/\text{mL}$ kanamycin. After overnight incubation, the bacterial suspensions were diluted 1:100 in LB media. Cells were grown at 37°C until the OD_{600} reached 0.3, then the cells were diluted 1:1000 in LB medium supplied with or without inducers (1% L-arabinose and 0.5 mM IPTG), while 2% glucose was added in non-

induced medium to inhibit the protein expression. Cells were dispensed (150 μ L) into a 96-well plate. And optical density measurements at a wavelength of 600 nm were taken every 10 min using a BioTek Synergy H1 at 37 $^{\circ}$ C.

Fig. 1: Overall architecture of the apo DSR2 tetramer.

b Growth curves of *E. coli* cells producing DSR2 alone or together with the phage tail tube protein. Protein production was induced by the addition of 0.5 mM IPTG and 1% L-Ara. The *E. coli* cells without induction were used as control. Curves represent three independent experiments.

- References # 8 and # 16 are the same.

We thank the reviewer for pointing out the duplicates of the reference. We have corrected this error accordingly. Thank you!

- References # 19 and # 22 are the same.

We thank the reviewer for pointing out the duplicates of the reference. We have corrected this error accordingly. Thank you!

- When discussing Thoeris antiphage defense, reference should be added to Leavitt et al. (2022) Nature- “Viruses inhibit TIR gcADPR signalling to overcome bacterial defence” which shows that ThsB of Thoeris produces a 1”-3’gcADPR isomer which binds to the SLOG domain of ThsA and initiates SIR2 domain NADase activity. This work built upon the currently referenced manuscript by Ofir et al. (2021) Nature which originally posits a signaling molecule but did not confirm the identity.

We greatly appreciate the reviewer for the deeply understanding of the Thoeris antiphage defense system. As the reviewer suggested, we have added this reference (reference #13) in our manuscript, which is invaluable in enhancing the accuracy of our manuscript.

- Remove the ‘s’ from hydrolases. “Thus, our findings suggest that the tetrameric DSR2

should function as a tube-activated NAD⁺ hydrolases rather than an NAD⁺-dependent deacetylase.”

We thank the reviewer for pointing out this typo. The extra ‘s’ has been removed from hydrolases. Thank you.

REVIEWERS' COMMENTS

Reviewer #1 (Remarks to the Author):

The authors have addressed my concerns. I endorse the publication of this manuscript in Nature communications.

Reviewer #2 (Remarks to the Author):

Additional mutagenesis data supports the original conclusions and combined with text and minor figure edits, the manuscript is in good shape. This is a very clear and comprehensive structural study of the DSR2 antiphage defense system. I am satisfied with the response to reviewers and updates to the manuscript.